# MSRS: Evaluating Multi-Source Retrieval-Augmented Generation

**Rohan Phanse   Yijie Zhou   Kejian Shi   Wencai Zhang   Yixin Liu**
**Yilun Zhao   Arman Cohan**

Yale University

{rohan.phanse, yilun.zhao, arman.cohan}@yale.edu

## Abstract

Retrieval-augmented systems are typically evaluated in settings where information required to answer the query can be found within a single source or the answer is short-form or factoid-based. However, many real-world applications demand the ability to integrate and summarize information scattered across *multiple* sources, where no single source is sufficient to respond to the user's question. In such settings, the retrieval component of a RAG pipeline must recognize a variety of relevance signals, and the generation component must connect and synthesize information across multiple sources. We present a scalable framework for constructing evaluation benchmarks that challenge RAG systems to integrate information across distinct sources and generate long-form responses. Using our framework, we build two new benchmarks on Multi-Source Retrieval and Synthesis: MSRS-STORY and MSRS-MEET, representing narrative synthesis and summarization tasks, respectively, that require retrieval from large collections. Our extensive experiments with various RAG pipelines—including sparse and dense retrievers combined with frontier LLMs—reveal that generation quality is highly dependent on retrieval effectiveness, which varies greatly by task. While multi-source synthesis proves challenging even in an oracle retrieval setting, we find that reasoning models significantly outperform standard LLMs at this distinct step.

https://github.com/yale-nlp/MSRS

## 1 Introduction

Traditional Retrieval-Augmented Generation (RAG) systems are often applied in settings where the necessary information to answer a query is contained within a single retrieved source (Fan et al., 2024). However, many real-world applications often require capturing and synthesizing information scattered across many sources. In these scenarios, the required information is often distributed, complementary, or even contradictory across sources, demanding multi-document retrieval and reasoning capabilities (Tang & Yang, 2024; Lála et al., 2023; Asai et al., 2024b; Zhao et al., 2025; Su et al., 2024; Wang et al., 2025).

Specifically, the earlier testbeds of RAG systems, such as multi-hop QA (Yang et al., 2018; Khashabi et al., 2018; Trivedi et al., 2022) are limited to task settings with two-hop questions and short-form answers. More recent studies have introduced benchmarks with increased query complexities and reasoning requirements (Fan et al., 2024). However, they are mostly restricted to short answers (Tang & Yang, 2024) or rely on synthesizing information from a narrow set of source documents (Edge et al., 2024). We argue that open-domain multi-document summarization (MDS) is a more challenging and realistic testbed, as it requires RAG systems to retrieve and synthesize key information from multiple, complementary long-form documents. In addition, this setting specifically targets the generation capabilities of LLMs for comprehensive, long-form summaries, rather than short-form multi-hop question-answering (Trivedi et al., 2022; Tang & Yang, 2024).

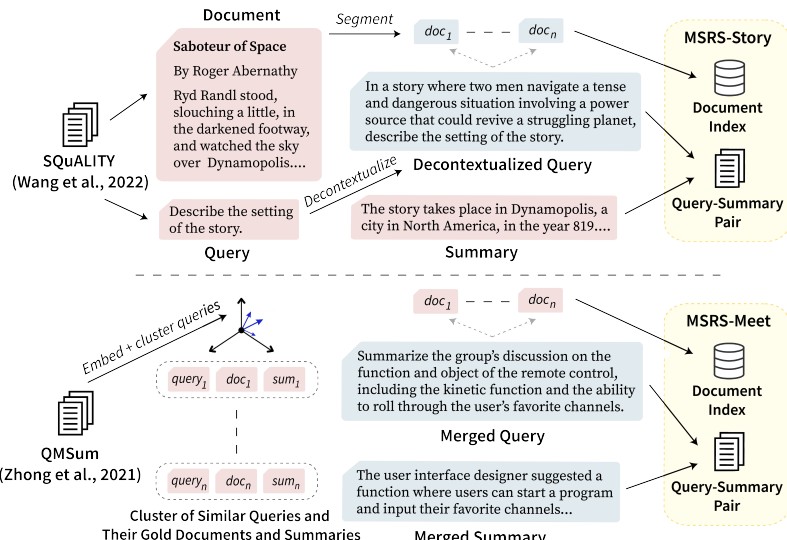

Figure 1: Overview of the creation process of MSRS-STORY and MSRS-MEET datasets.

Open-domain multi-document summarization (Liu et al., 2018; Giorgi et al., 2023) is a retrieval-based task, consisting of joint retrieval and summarization, where relevant documents to an information need or query are first retrieved, and then a summary of relevant information is generated and provided to the user. The source documents have to be retrieved from a large corpus of documents (as opposed to regular multi-document summarization (Fabbri et al., 2019) where the set of related input documents is already provided). In addition, such information may be disjoint, overlapping, or complementary between multiple documents.

While this task can be a suitable testbed for evaluating RAG systems in multi-source settings, progress is impeded by a lack of realistic benchmarks due to the difficulty of collecting data that includes both multiple complementary relevant documents to a given query and a corresponding ground-truth output. We aim to bridge this gap, particularly focusing on the limitations observed in recent work on open domain summarization (§2). Specifically, two primary limitations (Tang et al., 2021; Giorgi et al., 2023) include the use of unrealistic, pseudo-queries (using the reference summary of documents or document titles as the retrieval query) and the sub-optimal construction of the document corpus. To address these limitations, we propose a new method for data collection and introduce two new benchmarks for Multi-Source Retrieval and Synthesis: MSRS-STORY and MSRS-MEET (§3). Bootstrapped from existing long-context, query-focused summarization datasets, our benchmarks aim to meet the main requirements of a realistic multi-source RAG task: (1) necessitate retrieving and synthesizing information from a large corpus; (2) the information to address the query must come from multiple documents; and (3) the queries should be realistic, reflecting a user's real-world information need.

We experiment with a wide range of RAG pipelines, which include sparse and dense retrievers, and state-of-the-art LLMs as generators. The effectiveness of our tested retrievers varies significantly, highlighting the need for better solutions to address document quality and relevance variability challenges in complex RAG problems. Furthermore, the final generation performance is highly dependent on retrieval quality. The multi-source synthesis step remains challenging, even in an oracle setting of perfect retrieval, but achieves meaningful gains when using reasoning models over standard LLMs. Our contributions are summarized below:

- We propose a scalable approach (§3) to construct multi-source RAG datasets from query-focused MDS datasets.

| Task | Dataset | Domain | Query Source | # Examples | Avg. Summ. Len. (tokens) |
|---|---|---|---|---|---|
| MDS | DUC 2003-04 (Over et al., 2007) | News articles | - | 500 | 110 |
| | Multi-News (Fabbri et al., 2019) | News articles | - | 56,216 | 264 |
| | Multi-Xscience (Lu et al., 2020) | Scientific literature | - | 40,528 | 117 |
| | WCEP (Ghalandari et al., 2020) | News articles | - | 92,560 | 33 |
| | MS^2 (DeYoung et al., 2021) | Medical studies | - | 415,333 | 58 |
| Query-Focused MDS | DUC 2005-07 (Over et al., 2007) | News articles | Human written | 1,593 | 250 |
| | AQuaMuSe (Kulkarni et al., 2020) | Wikipedia | Rewritten from NQ dataset | 5,519 | 106 |
| | QMDSCNN (Pasunuru et al., 2021) | News articles | Title of news article | 371,971 | 250 |
| | QDSIR (Pasunuru et al., 2021) | General web | Bing search logs | 102,595 | 250 |
| | QMSum (Zhong et al., 2021) | Meeting transcripts | Human written | 1,808 | 70 |
| | SQuALITY (Wang et al., 2022) | Sci-fi stories | Human written | 625 | 237 |
| Open-Domain QA | ASQA (Stelmakh et al., 2022) | Wikipedia | NQ dataset & Wikipedia | 6,316 | 65 |
| | QuALITY (Pang et al., 2022) | Mixed-genre | Human written | 381 | 11 |
| Open-Domain MDS | **MSRS (ours)** | Meetings, Sci-fi stories | Rewritten from MDS dataset | 1,071 | 261 |

Table 1: Comparison between MSRS and other related datasets. # *Examples* refers to the number of unique sets of documents paired with a summary, and a query when applicable.

- We release MSRS-STORY and MSRS-MEET, two benchmarks designed to test RAG systems in retrieving complementary information from large data sources and providing a coherent answer to the user query.
- We conduct extensive experiments on RAG system components, specifically retrieval (§4.1) and generation (§4.2), providing insights into their capabilities and limitations.

## 2 Related Work

**Query-Focused MDS and Retrieval-Augmented Generation.** Multi-document summarization (MDS) is a classic NLP task where models synthesize information from multiple input sources (Yasunaga et al., 2017; Liu et al., 2018; Fabbri et al., 2019; Liu & Lapata, 2019; Li et al., 2020; Lu et al., 2020; DeYoung et al., 2021). Query-Focused MDS (QMDS) specializes this task by generating summaries from given documents that address a specific user query (Kulkarni et al., 2020; Pasunuru et al., 2021). When QMDS operates in open domains (also known as Open-Domain MDS (Giorgi et al., 2023)), it requires first retrieving relevant information before summarization. This setting provides a natural testbed for RAG systems when answers span multiple documents. While similar to recent deep-research systems (OpenAI, 2025d), open-domain MDS typically addresses narrower information needs and produces concise summaries rather than long, detailed reports seen in the output of deep research systems.

High-quality QMDS datasets remain scarce. Existing benchmarks like AQuaMuSe (Kulkarni et al., 2020) and QMDSIR (Pasunuru et al., 2021) automatically retrieve documents through corpus mining or web search but lack gold-standard relevance labels. QMDSCNN (Pasunuru et al., 2021) repurposes existing datasets by treating titles as pseudo-queries and using four retrieved documents per query, though this approach cannot guarantee document relevance or authentic information needs. Other benchmarks (Giorgi et al., 2023) suffer from unrealistic queries—using ground-truth summaries instead of real-world questions—and sparse corpora that yield many irrelevant documents. In contrast, MSRS presents more significant challenges in information retrieval, necessitating models to first retrieve query-relevant documents from a large corpus with complementary information with respect to each other and related to the query. This setting closely aligns with real-world information-seeking scenarios, where users typically do not have a predetermined set of documents.

**LLMs for Retrieval-Augmented Generation.** Core RAG methods vary in their retrieval strategies—from sparse methods like BM25 (Robertson et al., 1994) to dense-vector approaches and hybrid configurations—with effectiveness depending on the interplay between retrieval metrics and integration mechanisms (Li et al., 2022; Zhao & Li, 2024). Retrieval indexes based on LLM embeddings have been found to outperform strong *retrieval* baselines such as BM25 and self-supervised dense retrieval methods (Brown et al., 2020). They are particularly effective in tasks that lack sufficient supervised data (Schick & Schütze, 2020; Winata et al., 2021; Bonifacio et al., 2022). More recently, LLM-based agentic systems with

retrieval (Asai et al., 2024a; Zheng et al., 2025; Shao et al., 2023; Hu et al., 2025; Singh et al., 2025) have attracted increasing attention. These systems show superior performance on complex open-domain tasks such as generating literature reviews and composing detailed reports. However, our work focuses on in-depth evaluation of the performance of retrievers and LLMs, as they are the core components of more complex agentic systems. Our benchmark involves a focused task and a more controlled output, rather than detailed, multi-page output formats of deep research systems (OpenAI, 2025d). Our benchmark provides a systematic evaluation framework that isolates and measures the individual contributions of retrieval quality and generation accuracy, enabling researchers to identify specific bottlenecks in multi-source RAG pipelines.

## 3   MSRS Benchmark

A primary principle behind our benchmark is to require multiple complementary documents for generating a long-form response to a user's query. To do so, we construct our RAG benchmark sourced from existing high-quality benchmarks for long-context single-document summarization. We provide both gold documents and gold summaries, which allows our benchmark to measure both retrieval and generation quality of RAG systems. Below we detail the dataset construction pipeline:

### 3.1   Source Data Collection

We use *SQuALITY* (Wang et al., 2022) and *QMSum* (Zhong et al., 2021) as source datasets. *SQuALITY* (Wang et al., 2022) focuses on question-based summarization of long stories. Expert contractors wrote the summaries in *SQuALITY*, and in order to ensure quality through rigorous cross-annotator reviews, the summary for each question was written by four different annotators. The *SQuALITY* dataset features 127 stories with 635 questions, leading to a total of 2,540 summaries. *QMSum* (Zhong et al., 2021) is designed for query-based summarization of multi-domain long meeting transcripts. The summaries are derived from transcribed meetings, focusing on the extraction and summarization of relevant segments in response to specific queries. The dataset comprises 1,808 query-summary pairs that span across 232 meetings from multiple domains. These pairs were carefully created to cover a wide variety of queries and corresponding summaries.

These two source datasets are appropriate for our benchmark due to their high-quality annotations, diverse query types, and long-form content that necessitates multi-document reasoning. The queries in both datasets represent realistic information needs that require synthesizing information across multiple text segments. Using these two data sources, we create two new RAG benchmarks: MSRS-STORY and MSRS-MEET. Table 2 outlines the data statistics of each benchmark. We next detail the benchmark construction process.

### 3.2   MSRS-STORY Construction

Stories inherently chronicle a sequence of events, and chapters serve as natural boundaries that divide these events into individual documents. We segment the stories into chapters using natural web delimiters like `<hr class="chap"/>` in the HTML-scraped forms of the stories. These separate chapters are then considered as individual documents in an open-domain MDS setup. The questions in *SQuALITY* often pertain to multiple chapters of a story, inherently positioning our dataset as a information-seeking setup which is convenient for building a multi-document RAG dataset.

**Decontextualizing Queries.**   Original queries from *SQuALITY* are often context-dependent and vague when used standalone. To make them suitable for retrieval, they require decontextualization (Choi et al., 2021). We use GPT-4o mini[1] to rewrite these queries. Unlike approaches that generate new queries from documents and summaries (Laskar et al., 2023), our method modifies the existing query. This preserves the original question's exact intent and relevance to the gold summary, while making it self-contained for retrieval. Specifically, we prompted GPT-4o mini with the query and its corresponding story, asking it to append a

---

[1]We find GPT-4o mini (version: `gpt-4o-mini-2024-07-18`) to work well and be cost-effective.

| Dataset | Story | MEET |
|---|---|---|
| # Documents | 1,138 | 231 |
| Avg. Document Length (tokens) | 845.44 | 7,207.27 |
| Total Corpus Length (tokens) | 962,108 | 1,664,880 |
| # Queries | 635 | 436 |
| Avg. Query Length (tokens) | 26.94 | 32.89 |
| # Ref. Summaries per Query | 4 | 1 |
| Avg. Summary Length (tokens) | 273.80 | 185.17 |
| Training Set Size | 250 | 261 |
| Test Set Size | 260 | 131 |
| Development Set Size | 125 | 44 |

Table 2: Data statistics of the constructed MSRS-STORY and MSRS-MEET datasets.

| Annotation Quality | %S = 5 |
|---|---|
| Summary Coherence and Fluency | 95.6 |
| Factual Consistency and Accuracy | 89.4 |
| Open-Domain Retrieval Necessity | 87.5 |
| Multi-Doc Information Aggregation | 86.9 |
| Query Relevance and Coverage | 86.3 |
| Inter-Doc Relationship Capture | 83.1 |

Table 3: Human evaluation over 160 samples (80 from each subset). Two expert annotators rate the samples (1–5). Detailed guidelines are provided in Appendix §E.

brief, vague story description. A human annotator then iteratively refined the output by rerunning the generation process as needed, ensuring the rewritten query maintained its original intent and included vague retrieval clues—akin to the TREC Tip-of-the-Tongue task (Arguello et al., 2024)—without revealing the answer. The prompt used is in Appendix §C.2, and an example is provided in Appendix §F. We also empirically validate the importance of the decontextualization step in Appendix §B.3, showing that it enhances both retrieval and generation performance.

### 3.3 MSRS-MEET Construction

For *QMSum*, we leverage the dataset's thematic continuity and focused discussion characteristic. Each source document in *QMSum* is a transcript of a meeting, and the meetings typically revolve around persistent themes as participants strive to make progress in discussions. For instance, in the Product setting that includes 137 meetings about designing a new remote control, the topic of *buttons* recurs in 131 of them, *rubber buttons* in 24, and *colored buttons* in 3. When seeking information on *colored buttons*, an ideal scenario would involve a query specifically about *colored buttons*, with a summary consolidating discussions from the three relevant meetings.

**Cluster and Merge Query-Summary Pairs.** We cluster the original queries in the dataset based on their similarity, subsequently merging similar queries into a single, more comprehensive one. This process extends to summaries, merging them into a single summary that provides a cohesive overview of several meetings. The result is a query-summary pair naturally aligned with multiple meetings, providing a rich resource for open-domain MDS tasks. Details of the process and particular examples can be found in Appendix §A.

### 3.4 Human Validation of Data Quality

To ensure the high quality and utility of the MSRS, we conduct rigorous human evaluations. We randomly select 80 examples from each of the MSRS-MEET and MSRS-STORY subsets and ask two expert annotators to assess dataset examples on a scale of 1 to 5 using six criteria. The full annotation criteria and guidelines are presented in Appendix §E. As shown in Table 3, the vast majority of examples received high scores across all criteria. Over 80% of the examples were rated >4 for each criterion, with Summary Coherence and Fluency, and Open-Domain Retrieval Necessity scoring 95.6% and 87.5%, respectively. These results corroborate the suitability of MSRS for benchmarking open-domain MDS systems.

## 4 Experiment Setup

*Retrieve-then-summarize* (Liu et al., 2018; Giorgi et al., 2023) is a standard RAG process where relevant documents are *retrieved* from a large corpus using a query and a model generates the output by summarizing the information from retrieved sources that are relevant to the query. We implement and evaluate this pipeline for the MSRS tasks.

### 4.1 Retrieval Models

We incorporate both *sparse* and *dense* retrievers, along with *LLM-based reranking*, into our retrieval framework.

**Sparse Retrievers.** Sparse retrievers compute the relevance of a document to a query based on the frequency of overlapping terms. We use BM25 (Robertson et al., 1994) as a representative and widely used approach in this family.

**Dense Retrievers.** Dense retrievers embed both documents and queries in a shared embedding space. This provides a more flexible and context-aware approach to measuring the relatedness of text strings. We employed multiple different dense retrievers for our experiments: **NV-Embed** (Lee et al., 2025), **gte-Qwen2-instruct** (Li et al., 2023), **GritLM-7B** (Muennighoff et al., 2024), **Promptriever** (Weller et al., 2025), OpenAI's text-embedding-3 models,[2] and Google's gemini-embedding-exp-03-07. [3] We provide retrieval implementation details in Appendix §D.1.

**LLM-Based Reranking.** We performed LLM-based reranking for both sparse and dense retrievers to investigate whether the addition of a reranker could achieve additional performance gains. We used Gemini 2.0 Flash to rerank the top 20 retrieved documents for each query.[4] For each retrieved document, a pointwise relevance score between the document and query was generated using an LLM in a process similar to G-EVAL (Liu et al., 2023). We opted to use pointwise reranking instead of approaches that consider multiple documents at once, such as listwise and pairwise reranking, due to the substantial lengths of the documents. The reranking prompts are provided in Appendix §C.4.

### 4.2 Generation Models

We conduct experiments with a series of non-reasoning LLMs, namely **GPT-4o** (OpenAI, 2023), **Llama 2** (Touvron et al., 2023), **Llama 3.1** and **Llama 3.3** (Grattafiori et al., 2024), **Qwen2.5** (Yang et al., 2024), **Gemini 1.5 and 2.0** (Gemini Team, 2024), and **DeepSeek-V3** (DeepSeek-AI, 2024). We also evaluate reasoning LLMs, including **GPT-5** (OpenAI, 2025b), **o3** (OpenAI, 2025c), **gpt-oss** (OpenAI, 2025a), **Gemini 2.5** (Gemini Team, 2025), and **DeepSeek-R1** (DeepSeek-AI, 2025). Exact prompts and inference details are in Appendix §C and §D.2, respectively.

**Retrieval Evaluation.** We report NDCG (Järvelin & Kekäläinen, 2000), MAP (Voorhees & Harman, 1999), Precision@K, and Recall@K as standard IR evaluation metrics, with further details provided in Appendix §D.3.1.

**Generation Evaluation.** We evaluate the quality of summaries generated by different models using the following three metrics commonly used to evaluate model-generated output against a gold standard (see Appendix §D.3.2 for details): **ROUGE** (Lin, 2004), **BERTScore** (Zhang et al., 2020), and LLM-as-a-judge with **G-EVAL** (Liu et al., 2023), which employs LLMs with chain-of-thought (CoT) prompting and rubrics to assess the quality of outputs. We compute G-EVAL scores using the relevance rubric (Appendix §C.5) as it is the only G-EVAL rubric that directly assesses semantic similarity between texts. We provide our G-Eval prompts in Appendix §C.5 and detail our implementation in Appendix §D.3.2.

## 5 Experimental Result and Analysis

In this section, we present our findings. We first analyze the performance of various retrieval methods (§5.1) and then evaluate the final generation quality (§5.2), including specific analyses on data contamination, the necessity of multiple documents, and the capabilities of long-context models. Finally, we examine the sources of error in the oracle setting (§5.3) and assess the performance of reasoning models in this setting (§5.4).

---

[2]https://platform.openai.com/docs/guides/embeddings
[3]https://ai.google.dev/gemini-api/docs/embeddings
[4]We did not experiment with stronger LLMs as rerankers due to their higher cost. Gemini 2.0 Flash provides a reasonable balance of performance and efficiency.

| Method | | | MSRS-STORY | | | |
| --- | --- | --- | --- | --- | --- | --- |
| | P@8 | R@8 | NDCG | MAP | G-EVAL #1 | G-EVAL #2 |
| BM25 | 20.91 | 23.20 | 28.75 | 18.33 | 38.54 | 42.90 |
| BM25 + Rerank | 27.45 | 30.10 | 40.04 | 27.41 | 41.92 | 47.15 |
| Promptriever | 45.62 | 45.09 | 58.46 | 40.68 | 46.82 | 56.99 |
| gte-Qwen2-1.5B-instruct | 45.96 | 46.97 | 59.15 | 42.49 | 46.56 | 56.45 |
| gte-Qwen2-1.5B-instruct + Rerank | 50.38 | 50.65 | 63.57 | 45.26 | 48.41 | 57.79 |
| gte-Qwen2-7B-instruct | 50.67 | 51.21 | 63.95 | 45.02 | 48.12 | 57.28 |
| GritLM-7B | 51.01 | 50.09 | 64.09 | 45.51 | 49.12 | 57.40 |
| NV-Embed-v1 | 45.19 | 43.11 | 56.63 | 38.27 | 46.41 | 55.40 |
| NV-Embed-v2 | 51.49 | 49.81 | 63.40 | 44.91 | 48.08 | 57.01 |
| NV-Embed-v2 + Rerank | 52.74 | 51.08 | 64.38 | 45.03 | 48.35 | 57.15 |
| text-embedding-3-small | 41.73 | 43.15 | 54.53 | 37.35 | 46.95 | 55.09 |
| text-embedding-3-large | 46.59 | 46.76 | 58.51 | 41.25 | 46.53 | 55.64 |
| gemini-embedding | **56.68** | **57.46** | **72.33** | **54.25** | **50.67** | 59.54 |
| gemini-embedding + Rerank | 55.48 | 55.89 | 69.40 | 50.29 | 50.09 | **59.65** |

Table 4: Retrieval performance for MSRS-STORY. G-EVAL #1 and G-EVAL #2 refer to the G-EVAL scores for the summaries produced by Qwen2.5-7B and Gemini 2.0 Flash, respectively.

## 5.1 Retrieval Results

The retrieval performance results for various methods on MSRS-STORY and MSRS-MEET are presented in Table 4 and Table 5, respectively. These tables report standard IR metrics as well as downstream generation quality (G-EVAL) using summaries generated from the retrieved documents.

For MSRS-STORY (Table 4), dense retrieval methods show clear advantages over the sparse BM25 baseline. BM25 achieves an NDCG of 28.75, whereas all dense retrievers surpass this significantly. Gemini-embedding consistently performs best across all standard IR metrics (e.g., NDCG 72.33, MAP 54.25). LLM-based reranking generally provides performance gains, particularly for BM25 (NDCG +11.29 points) and mid-range dense retrievers like gte-Qwen2-1.5B-instruct (NDCG +4.42 points). However, the benefit diminishes for the top-performing dense models; reranking NV-Embed-v2 yields only a modest gain, and reranking gemini-embedding results in slightly lower IR scores but a marginal improvement in one of the G-EVAL scores. Overall, IR metrics and downstream G-EVAL scores appear reasonably correlated on MSRS-STORY.

MSRS-MEET (Table 5) appears significantly more challenging (best NDCG score is 32.80). Notably, BM25 (NDCG 27.58) is competitive, outperforming several dense retrievers like Promptriever and GritLM-7B, and performing comparably to others before reranking. Reranking consistently proves beneficial, improving scores across most methods. The top-performing system overall is NV-Embed-v2 + Rerank, achieving the best P@3, NDCG, and Gemini 2.0 Flash G-EVAL score. Unlike MSRS-STORY, the correlation between standard IR metrics and downstream G-EVAL scores is less pronounced on MSRS-MEET; for example, BM25 achieves respectable IR scores but the lowest G-EVAL scores, suggesting that simple relevance metrics may not fully capture the nuances required for effective multi-document synthesis in this domain. The significant performance variation across methods and datasets underscores the challenges inherent in multi-source retrieval for RAG.

## 5.2 Final Generation Results

The performance of summarizers is presented in Table 6 and Table 7. For both datasets, there is a strong correlation between retrieval quality and generation quality. As retrieval performance improves (e.g., from BM25 to stronger dense retrievers), the downstream summarization scores (G-EVAL and ROUGE-2) generally increase for all models. For instance, on MSRS-STORY, the average G-EVAL score across all models rises from 40.53 with BM25 retrieval to 53.34 with gemini-embedding retrieval, approaching the oracle average of

| | | | MSRS-MEET | | | |
|---|---|---|---|---|---|---|
| **Method** | **P@3** | **R@3** | **NDCG** | **MAP** | **G-EVAL #1** | **G-EVAL #2** |
| BM25 | 18.32 | 26.08 | 27.58 | 21.56 | 39.29 | 40.93 |
| BM25 + Rerank | 21.63 | 30.84 | 32.32 | 25.18 | 43.19 | 43.59 |
| Promptriever | 12.47 | 18.36 | 19.32 | 15.03 | 40.29 | 42.71 |
| GritLM-7B | 15.01 | 20.67 | 22.02 | 16.02 | 41.02 | 42.83 |
| gte-Qwen2-1.5B-instruct | 21.37 | 28.72 | 31.60 | 24.21 | 41.19 | 43.41 |
| gte-Qwen2-1.5B-instruct + Rerank | 21.37 | 30.46 | 31.53 | 24.19 | 43.35 | 42.38 |
| gte-Qwen2-7B-instruct | 18.83 | 26.28 | 26.16 | 19.61 | 42.80 | 44.57 |
| NV-Embed-v1 | 18.07 | 25.60 | 26.83 | 20.70 | 42.43 | 43.21 |
| NV-Embed-v2 | 20.10 | 27.93 | 31.02 | 23.93 | 41.53 | 41.77 |
| NV-Embed-v2 + Rerank | **23.41** | 32.12 | **32.80** | 25.15 | 43.70 | **45.13** |
| text-embedding-3-small | 17.30 | 25.06 | 26.36 | 20.43 | 41.12 | 42.45 |
| text-embedding-3-large | 18.58 | 25.71 | 26.49 | 20.32 | 42.28 | 42.40 |
| gemini-embedding | 21.63 | 29.61 | 31.04 | 23.84 | 41.56 | 42.35 |
| gemini-embedding + Rerank | 21.88 | **32.35** | 32.10 | **25.28** | **44.06** | 44.02 |

Table 5: Retrieval performance for MSRS-MEET. G-EVAL #1 and G-EVAL #2 refer to the G-EVAL scores for the summaries produced by Qwen2.5-7B and Gemini 2.0 Flash, respectively.

| | | | | | MSRS-STORY | | | | | | |
|---|---|---|---|---|---|---|---|---|---|---|---|
| | **BM25** | | **gte-Qwen2-1.5B** | | **NV-Embed-v2** | | **gemini-emb** | | **Average** | | **Oracle** | |
| | R-2 | G-EVAL | R-2 | G-EVAL | R-2 | G-EVAL | R-2 | G-EVAL | R-2 | G-EVAL | R-2 | G-EVAL |
| Llama 2 - 7B | 7.17 | 31.13 | 7.98 | 36.38 | 8.02 | 37.58 | 8.17 | 39.40 | 7.84 | 36.12 | 8.97 | 45.13 |
| Llama 2 - 70B | 7.53 | 32.66 | 8.39 | 39.97 | 8.71 | 40.58 | 8.97 | 42.24 | 8.40 | 38.86 | 9.47 | 48.95 |
| Llama 3.1 - 8B | 7.42 | 37.01 | 8.07 | 44.85 | 8.35 | 46.25 | 8.72 | 48.79 | 8.14 | 44.23 | 9.19 | 55.77 |
| Qwen2.5 - 7B | 6.86 | 38.54 | 7.75 | 46.56 | 8.13 | 48.08 | 8.04 | 50.67 | 7.70 | 45.96 | 9.07 | 57.40 |
| Llama 3.3 - 70B | 7.52 | 40.88 | 8.45 | 49.02 | 8.20 | 50.27 | 8.95 | 51.22 | 8.28 | 47.85 | 9.80 | 58.88 |
| Llama 3.1 - 70B | 7.99 | 41.20 | 9.13 | 50.41 | 9.29 | 50.89 | 9.22 | 51.29 | 8.91 | 48.45 | 10.49 | 59.85 |
| Qwen2.5 - 72B | 7.74 | 43.95 | 8.71 | 54.78 | 8.87 | 55.56 | 9.06 | 59.20 | 8.60 | 53.37 | 10.45 | 67.70 |
| GPT-4o mini | 7.17 | 44.58 | 8.04 | 51.75 | 8.08 | 54.14 | 8.44 | 56.67 | 7.93 | 51.78 | 9.19 | 61.08 |
| GPT-4o | 7.92 | 45.86 | 9.20 | 55.88 | 9.14 | 57.48 | 9.46 | 59.88 | 8.93 | 54.77 | 10.70 | 68.22 |
| Gemini 1.5 Pro | 7.21 | 40.92 | 8.88 | 55.38 | 9.01 | 55.71 | 9.83 | 59.53 | 8.73 | 52.89 | 10.79 | 68.69 |
| Gemini 2.0 Flash | 8.07 | 42.90 | **9.87** | 56.45 | **9.81** | 57.01 | **10.47** | 59.54 | **9.55** | 53.97 | **11.64** | 69.22 |
| DeepSeek-V3 | **8.29** | **46.69** | 9.39 | **57.72** | 9.49 | **58.49** | 10.00 | **61.63** | 9.29 | **56.13** | 10.85 | **69.85** |
| Average | 7.57 | 40.53 | 8.66 | 49.93 | 8.76 | 51.00 | 9.11 | 53.34 | 8.53 | 48.70 | 10.05 | 60.90 |

Table 6: Final generation performance on MSRS-STORY. R-2 refers to the ROUGE-2 F1 score. The performance with different retrievers and the oracle document set is reported.

60.90 (Table 6). On MSRS-MEET, reranking BM25 or NV-Embed-v2 results in an average G-EVAL improvement across models (Table 7).

For MSRS-STORY, we observe larger models generally yield better summaries (Table 6). The top-performing models include DeepSeek-V3, Gemini 2.0 Flash, and GPT-4o, particularly when coupled with high-quality retrieval like gemini-embedding. DeepSeek-V3 achieves the highest G-EVAL scores across all standard retrieval settings and oracle setting.

On MSRS-MEET (Table 7), overall generation scores are lower, but similar model capability trends hold. GPT-4o mini shows strong results, achieving the highest G-EVAL scores with BM25 (45.29), BM25 + Rerank (47.13), and NV-Embed-v2 + Rerank (47.07). GPT-4o performs best with the NV-Embed-v2 retriever (45.73) and in the oracle setting (53.67). These results highlight that while advanced generators are crucial, their effectiveness is heavily dependent on the quality of the retrieved documents, which highlights the importance of both components in the RAG pipeline for complex multi-document tasks.

| | BM25 | | BM25 Rerank | | NV-Embed-v2 | | NV2 Rerank | | Average | | Oracle | |
|---|---|---|---|---|---|---|---|---|---|---|---|---|
| | R-2 | G-EVAL | R-2 | G-EVAL | R-2 | G-EVAL | R-2 | G-EVAL | R-2 | G-EVAL | R-2 | G-EVAL |
| Llama 2 - 7B | 7.60 | 36.03 | 8.18 | 36.15 | 8.27 | 37.84 | 8.45 | 38.40 | 8.12 | 37.11 | 8.87 | 41.12 |
| Llama 2 - 70B | 8.70 | 37.15 | 9.04 | 39.68 | 8.81 | 40.30 | 8.82 | 39.89 | 8.84 | 39.25 | 9.75 | 42.21 |
| Llama 3.1 - 8B | 6.88 | 37.20 | 7.70 | 41.42 | 6.86 | 39.67 | 7.25 | 41.22 | 7.17 | 39.88 | 7.62 | 44.29 |
| Llama 3.1 - 70B | 8.18 | 39.53 | 8.35 | 41.25 | 8.54 | 40.74 | 8.74 | 43.14 | 8.45 | 41.17 | 9.52 | 45.46 |
| Llama 3.3 - 70B | 7.16 | 40.04 | 7.89 | 43.55 | 8.13 | 43.04 | 7.93 | 43.06 | 7.78 | 42.42 | 8.80 | 47.11 |
| Qwen2.5 - 7B | 7.93 | 39.29 | 7.81 | 43.19 | 8.12 | 41.53 | 8.14 | 43.70 | 8.00 | 41.93 | 8.83 | 48.87 |
| Qwen2.5 - 72B | 8.71 | 43.21 | 9.20 | 46.01 | 8.88 | 44.26 | 9.17 | 46.47 | 8.99 | 44.99 | 10.38 | 53.25 |
| Gemini 1.5 Pro | 7.89 | 37.54 | 8.03 | 40.86 | 8.01 | 39.98 | 8.10 | 40.98 | 8.01 | 39.84 | 9.58 | 47.97 |
| Gemini 2.0 Flash | 8.99 | 40.93 | 9.10 | 43.59 | 8.77 | 41.77 | 9.35 | 45.13 | 9.05 | 42.86 | **10.71** | 52.31 |
| DeepSeek-V3 | **9.04** | 44.23 | **9.44** | 45.95 | **9.25** | 44.97 | 8.97 | 46.53 | 9.17 | 45.42 | 10.29 | 53.52 |
| GPT-4o mini | 8.71 | **45.29** | 9.06 | **47.13** | 8.90 | 45.66 | 8.97 | **47.07** | 8.91 | **46.29** | 9.96 | 53.56 |
| GPT-4o | 8.72 | 45.08 | 9.22 | 46.57 | 9.19 | **45.73** | 9.58 | 46.85 | 9.18 | 46.06 | 10.64 | **53.67** |
| Average | 8.21 | 40.46 | 8.58 | 42.95 | 8.48 | 42.12 | 8.62 | 43.54 | 8.47 | 42.27 | 9.58 | 48.61 |

Table 7: Final generation performance on MSRS-MEET. R-2 refers to the ROUGE-2 F1 score.

**Data Contamination Evaluation.** To quantify the level of potential data contamination, we construct the *No Documents* retrieval setting in which models are prompted to answer queries by relying on pre-existing knowledge without access to documents. For the queries in MSRS-STORY, the models were given the title and author of the relevant story along with a mention of the source dataset *SQuALITY*. For the queries in MSRS-MEET, the models were prompted to recall the meeting transcripts in *QMSum* (prompts in Appendix §C.6). The results (Table 9 in Appendix B.1) show that, without documents, models perform significantly worse than the oracle setting.

**Necessity of Multi-Document Retrieval.** To test the necessity of having multiple relevant documents during summarization, we create the *Oracle - Top 1* retrieval setting. For each query and its set of oracle documents, GPT-4o was used to score each oracle document based on relevance to the query with only the highest-scoring document being chosen (prompts in Appendix §C.3). Table 9 in Appendix §B.1 shows that the models achieve noticeable G-EVAL performance gains in MSRS-STORY when given all oracle documents instead of only the top one. For MSRS-MEET, the gaps in G-EVAL scores are smaller but still meaningful.

**Long-Context Evaluation.** Given that certain frontier LLMs can process very long contexts, we were interested in studying the extent to which retrieval is necessary if we can fit the entire corpus in an LLM's context. To this end, we assessed performance using Gemini 2.5 Pro (1M-token context) directly, providing the query and all source documents without explicit retrieval. While no truncation was needed for MSRS-STORY, documents in MSRS-MEET were individually truncated to fit the 1.6M-token corpus (Table 2) within the context limit (more details in Appendix §B.2). Table 10 in Appendix §B.2 shows that for MSRS-STORY, Gemini 2.5 Pro's long-context performance (R-2 8.91, G-EVAL 61.30) is competitive but surpassed by its performance with a strong retriever (R-2 9.63, G-EVAL 70.63) and with the oracle document set (R-2 10.43, G-EVAL 76.66). A similar trend is observed for MSRS-MEET in Table 10, indicating that while long-context models are promising, strong retrieval results in better performance, with further gains possible in the oracle setting.

## 5.3 Oracle Error Analysis

To investigate the source of errors in the oracle setting, where retrieval errors are absent, we conducted a human evaluation of the summaries produced by the top-performing models in Tables 6 and 7 in the oracle setting: DeepSeek-V3 for MSRS-STORY (G-EVAL 69.85) and GPT-4o for MSRS-MEET (G-EVAL 53.67); see Appendix §B.4 for more details. As shown in Table 12 in Appendix §B.4, summaries often omitted multiple minor details (40% for MSRS-STORY and 60% for MSRS-MEET), with MSRS-MEET also missing multiple major

| | MSRS-STORY- Oracle | | | | MSRS-MEET- Oracle | | | |
|---|---|---|---|---|---|---|---|---|
| **Method** | **R-1** | **R-2** | **BScr** | **G-EVAL** | **R-1** | **R-2** | **BScr** | **G-EVAL** |
| GPT-4o | 41.95 | 10.70 | 85.57 | 68.22 | 41.08 | **10.64** | 86.02 | 53.67 |
| DeepSeek-V3 | 42.08 | **10.85** | **85.96** | 69.85 | **41.16** | 10.29 | **86.13** | 53.52 |
| gpt-oss-20b | 40.81 | 9.63 | 84.67 | 64.51 | 36.55 | 7.68 | 83.87 | 54.31 |
| GPT-5 nano | 40.55 | 9.17 | 84.61 | 70.19 | 35.45 | 7.31 | 84.21 | 57.01 |
| gpt-oss-120b | 40.21 | 9.06 | 84.21 | 70.70 | 35.10 | 7.49 | 83.30 | 58.41 |
| DeepSeek-R1-0528 | 38.85 | 8.73 | 84.85 | 73.09 | 35.49 | 7.26 | 84.39 | 55.86 |
| Gemini 2.5 Flash | 41.40 | 10.07 | 85.39 | 76.05 | 40.82 | 9.66 | 85.45 | 57.98 |
| Gemini 2.5 Pro | **42.56** | 10.43 | 85.46 | 76.66 | 40.75 | 9.30 | 85.61 | 55.99 |
| o3 | 40.18 | 8.03 | 84.22 | 78.18 | 33.82 | 5.57 | 83.43 | 56.16 |
| GPT-5 mini | 39.91 | 8.75 | 84.45 | 78.95 | 33.42 | 5.91 | 83.61 | 60.40 |
| GPT-5 | 40.62 | 8.71 | 84.35 | **81.07** | 34.32 | 6.09 | 83.58 | **60.78** |

Table 8: Final generation performance on MSRS-STORY and MSRS-MEET for reasoning models given the oracle document set. The default reasoning effort level was used: "medium" for OpenAI models and "dynamic thinking" for Gemini models. R-1, R-2, and BScr refer to ROUGE-1, ROUGE-2, and BERTScore F1 scores, respectively. The DeepSeek-V3 and GPT-4o results (Table 6 and Table 7) are included as top non-reasoning model baselines.

details (35% of the time) and other error types appearing less frequently. These findings align with prior work (Hosseini et al., 2025; Kurisinkel & Chen, 2023; Tang et al., 2025) highlighting the challenges that even frontier LLMs face in multi-document tasks. The greater difficulty of MSRS-MEET is further supported by recent evaluations (Mullick et al., 2024; Chen et al., 2025) upon the *QMSum* benchmark (Zhong et al., 2021).

## 5.4 Reasoning Model Generation Results

Following the oracle error analysis (§5.3), which found that even top non-reasoning models missed key details, we evaluated the generation performance of state-of-the-art reasoning LLMs in the oracle setting. As shown in Table 8, these models achieve substantial gains in G-EVAL over the best non-reasoning baselines for both MSRS-STORY (up to +11.22) and MSRS-MEET (up to +7.11). Within these models, GPT-5 achieved the highest G-EVAL for both MSRS-STORY (81.07) and MSRS-MEET (60.78). We also observe that higher G-EVAL scores for reasoning models are often accompanied by lower ROUGE-2 and BERTScore values, suggesting that their improvements stem from generating more abstractive and less extractive summaries. Overall, these results demonstrate that reasoning models are effective for the complex synthesis tasks in MSRS, especially compared to non-reasoning models.

## 6 Conclusion

We introduce MSRS (MSRS-STORY and MSRS-MEET), novel benchmarks designed for multi-source retrieval-augmented generation tasks, pushing RAG evaluation beyond simpler settings. Built via a scalable, human-validated methodology, MSRS requires synthesizing information from multiple complementary documents for realistic queries. Our comprehensive experiments, analyzing diverse RAG pipelines (sparse/dense retrievers, state-of-the-art LLMs), revealed significant variations and challenges in multi-source retrieval, with effectiveness directly impacting generation quality, as captured by metrics like G-EVAL. Furthermore, we find that reasoning models achieve significant improvements over standard LLMs across MSRS, demonstrating their effectiveness for complex synthesis tasks. Further analyses confirmed the necessity of multi-document reasoning in MSRS. By providing a challenging and realistic testbed, MSRS identifies critical retrieval bottlenecks and serves as a valuable resource to drive advancements in RAG systems for complex information synthesis.

## Limitations and Future Work

Despite efforts to ensure quality, the complexity and scale of our dataset, combined with the use of automated query construction, may introduce minor inaccuracies but the nature does not undermine the value of our work. The inclusion of a large number of documents aimed to simulate realistic open-domain settings inevitably introduces a level of noise and that poses a challenge in evaluating the multi-source RAG systems. Our evaluation setup focuses on standard RAG pipelines with single-call LLMs as opposed to more complex iterative or argentic retrieval setups. Advanced methods such as query decomposition, retrieval planning, and multi-stage synthesis may help improve performance on our benchmarks.

## Acknowledgments

This work was supported in part by a research award from Meta, and compute credits from Google's TRC program. We thank Yale NLP lab members for helpful discussions.

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

# Appendix

## A  Transforming QMSum

We formalize the process of transforming *QMSum* into a new dataset as follows:

- **Meeting Transcripts:** Let $T = \{T_1, T_2, \ldots, T_n\}$ denote the set of all meeting transcripts, where $T_i$ represents the transcript of the $i$-th meeting.

- **Queries and Summaries:** Each transcript $T_i$ has an associated set of queries $Q_i = \{q_{i1}, q_{i2}, \ldots, q_{im}\}$, where $q_{ij}$ is the $j$-th query in the $i$-th meeting. Each query $q_{ij}$ has a corresponding summary $s_{ij}$, giving a set of summaries $S_i = \{s_{i1}, s_{i2}, \ldots, s_{im}\}$ for each transcript $T_i$.

- **Query Clustering:** Let $\text{sim}(q_{ij}, q_{kl})$ denote a function measuring similarity between two queries $q_{ij}$ and $q_{kl}$ from meetings $i$ and $k$, respectively. We implement $\text{sim}(q_{ij}, q_{kl})$ using `text-embedding-ada-002`[5]. A threshold $\theta$ is chosen such that if $\text{sim}(q_{ij}, q_{kl}) > \theta$, the queries are placed in the same cluster. This step produces a set of query clusters $C = \{C_1, C_2, \ldots, C_p\}$.

- **Cluster Modification:** Each cluster is constrained to a minimum and maximum size to ensure clusters are meaningful (not too small) and manageable (not too large). We chose *min* to be 2 and *max* to be 6. For any cluster $C_k$ containing greater than *max* queries, we apply a secondary clustering step within $C_k$ using a lower similarity threshold. Conversely, for any cluster $C_k$ containing fewer than *min* queries, we merge it with the most similar cluster. After these adjustments, we obtain a modified set of clusters $C' = \{C'_1, C'_2, \ldots, C'_q\}$.

- **Merging:** For each modified cluster $C'_k$, a merged query $Q'_k$ and a corresponding summary $S'_k$ are created by combining the queries and summaries in the cluster. This merging is performed by an LLM. The result is a set of merged query-summary pairs $\{(Q'_1, S'_1), (Q'_2, S'_2), \ldots, (Q'_q, S'_q)\}$, with each pair corresponding to a modified cluster.

The final outcome of this process is a new set of query-summary pairs $\{(Q_k, S_k)\}$, each providing a cohesive overview of multiple meetings.

We show one particular example here.

Query list:

- What did the group discuss about the function of rolling through the user's favorite channels?

- What did the group discuss about the object of the remote control?

- What did the group discuss about the functional Remote control?

- What function did the group think should be on the remote control?

- What did the group discuss about the functions of the remote control?

- How did they discuss the kinetic function of the remote control?

Result:

- Summarize the group's discussion on the function and object of the remote control, including the kinetic function and the ability to roll through the user's favorite channels.

---

[5]https://platform.openai.com/docs/guides/embeddings

| | MSRS-STORY | | | | | | MSRS-MEET | | | | | |
|---|---|---|---|---|---|---|---|---|---|---|---|---|
| | No Docs. | | Oracle - Top 1 | | Oracle | | No Docs. | | Oracle - Top 1 | | Oracle | |
| | R-2 | G-EVAL | R-2 | G-EVAL | R-2 | G-EVAL | R-2 | G-EVAL | R-2 | G-EVAL | R-2 | G-EVAL |
| GPT-4o mini | 5.92 | 33.65 | 7.96 | 53.49 | 9.19 | 61.08 | 7.38 | 38.60 | 9.58 | 49.40 | 9.96 | 53.56 |
| Qwen2.5 - 72B | 6.11 | 29.79 | 7.98 | 53.59 | 10.45 | 67.70 | 6.81 | 38.11 | 9.39 | 49.11 | 10.38 | 53.25 |
| GPT-4o | **6.74** | **37.60** | 9.06 | 57.02 | 10.70 | 68.22 | **7.74** | 39.53 | 9.65 | 49.08 | 10.64 | **53.67** |
| Gemini 1.5 Pro | 5.73 | 27.58 | 8.55 | 55.33 | 10.79 | 68.69 | 6.27 | 35.57 | 8.68 | 43.65 | 9.58 | 49.97 |
| Gemini 2.0 Flash | 6.14 | 29.61 | **9.37** | 56.57 | **11.64** | 69.22 | 6.58 | 37.41 | 10.01 | 45.77 | **10.71** | 52.31 |
| DeepSeek-V3 | 6.55 | 33.40 | 8.96 | **57.41** | 10.85 | **69.85** | 7.66 | **39.56** | 10.24 | **49.70** | 10.29 | 53.52 |

Table 9: Generation performance on MSRS-STORY and MSRS-MEET for the retrieval settings used in the data contamination and multi-document necessity evaluations in Section 5.2.

| | MSRS-STORY- Gemini 2.5 Pro | | | | MSRS-MEET- Gemini 2.5 Pro | | | |
|---|---|---|---|---|---|---|---|---|
| **Method** | **R-1** | **R-2** | **BScr** | **G-EVAL** | **R-1** | **R-2** | **BScr** | **G-EVAL** |
| Long Context | 40.19 | 8.91 | 84.78 | 61.30 | 37.54 | 7.46 | 84.80 | 43.82 |
| Strong Retriever | 41.63 | 9.63 | 85.27 | 70.63 | 38.45 | 7.70 | 85.08 | 45.36 |
| Oracle | **42.56** | **10.43** | **85.46** | **76.66** | **40.75** | **9.30** | **85.61** | **55.99** |

Table 10: Generation performance on MSRS-STORY and MSRS-MEET for the retrieval settings used in the long-context evaluation in Section 5.2. *Strong Retriever* denotes a representative strong retrieval method from Table 4 and Table 5: gemini-embedding for MSRS-STORY and NV-Embed-v2 + Rerank for MSRS-MEET.

# B   Additional Results

## B.1   Data Contamination and Multi-Document Necessity Evaluations

We provide the results of our data contamination and multi-document necessity experiments from Section 5.2 in Table 9.

## B.2   Long-Context Evaluation

The results of our long-context experiments described in Section 5.2 are shown in Table 10. In these experiments, all documents were provided to Gemini 2.5 Pro (1M-token context) without explicit retrieval. For MSRS-STORY, the entire corpus (962,108 tokens; Table 2) fit within the model's context window. For MSRS-MEET, however, the corpus size (1,664,800 tokens; Table 2) exceeded this limit. To address this, we truncated each of the 231 documents in MSRS-MEET individually, retaining only the first 4,800 tokens. Eighty documents were already below the limit, while the remaining 151 required truncation.

## B.3   Query Decontextualization Evaluation

To determine the necessity of query decontextualization for MSRS-STORY, we analyzed the original *SQuALITY* queries that were decontextualized to create the 260-query test set for MSRS-STORY, and found that nearly a third lacked sufficient standalone information. Specifically, 52 queries were slight variations of *"What is the plot of the story?"*, and an additional 30 queries asked about the setting in similarly unspecified ways, such as *"What is the setting of the story?"* or *"Describe the setting of the story."*

To determine the impact of the decontextualization step on retrieval and downstream summarization performance, we reconducted two retrieval runs from Table 4 using the original queries from *SQuALITY* instead of the decontextualized queries. We chose to evaluate a sparse retriever (BM25) and a strong dense retriever (gemini-embedding) as representative models, and report their performance on both the original *SQuALITY* queries and the decontextualized MSRS-STORY queries in Table 11.

| | | MSRS-STORY | | | | | |
|---|---|---|---|---|---|---|---|
| | Retrieval Setting | P@8 | R@8 | NDCG | MAP | G-EVAL #1 | G-EVAL #2 |
| Decontextualized Queries | BM25 | **20.91** | **23.20** | **28.75** | **18.33** | **38.54** | **42.90** |
| Original SQuALITY Queries | BM25 | 19.04 | 19.16 | 24.49 | 16.60 | 33.56 | 36.84 |
| Decontextualized Queries | gemini-embedding | **56.68** | **57.46** | **72.33** | **54.25** | **50.67** | **59.54** |
| Original SQuALITY Queries | gemini-embedding | 34.52 | 34.91 | 43.86 | 32.11 | 39.68 | 46.89 |

Table 11: Retrieval performance for MSRS-STORY when using decontextualized queries or the original *SQuALITY* queries. G-EVAL #1 and G-EVAL #2 refer to the G-EVAL scores for the summaries produced by Qwen2.5-7B and Gemini 2.0 Flash, respectively, for each retrieval setting.

The results show a clear and consistent benefit from query decontextualization across all retrieval settings. With BM25, decontextualized queries outperform the original SQuALITY queries on all metrics, including a 5+ point gain in G-Eval scores, demonstrating that even sparse retrievers benefit from more self-contained queries.

The effect is even more pronounced with gemini-embedding, where retrieval metrics improve dramatically (e.g., NDCG increases from 43.86 to 72.33), and G-Eval scores improve by 11+ points. These results confirm that decontextualization enables more effective retrieval by making previously underspecified queries solvable and more suited to the retrieval needs of real-world RAG settings.

## B.4 Oracle Error Analysis Study

We conducted a small-scale error analysis study with the top-performing models in Tables 6 and 7 for the oracle setting. These models were DeepSeek-V3 for MSRS-STORY (G-EVAL 69.85) and GPT-4o for MSRS-MEET (G-EVAL 53.67).

For this study, forty summaries (twenty per subset) were randomly sampled from the pools of summaries generated by DeepSeek-V3 given the MSRS-STORY oracle documents and GPT-4o given the MSRS-MEET oracle documents. The evaluations were divided up equally among two annotators, who selected all the error type labels that applied to each generated summary given the query, the gold-standard summary, and the oracle documents as reference. The prevalence of the human-annotated error types is included in Table 12.

| | MSRS-STORY Oracle - DeepSeek-V3 | MSRS-MEET Oracle - GPT-4o |
|---|---|---|
| Error Type | Error Prevalence (%) | Error Prevalence (%) |
| Missing Many Minor, Fine Details | 40 | 60 |
| Missing Many Major, Critical Details | 0 | 35 |
| Minor Query Misunderstanding | 0 | 5 |
| Major Query Misunderstanding | 0 | 10 |
| Presence of Hallucinations | 10 | 10 |
| Excessively General or Vague Response | 0 | 10 |

Table 12: Human evaluation over 40 samples (20 from each subset) analyzing the source of errors in the oracle setting with top-performing models for MSRS-STORY and MSRS-MEET.

## B.5 Full Summarization Results

We provide the full summarization results for MSRS-STORY and MSRS-MEET in Table 13 and Table 14, respectively. These results expand upon those in Table 6 and Table 7 by including two additional metrics: ROUGE-1 and BERTScore.

| | BM25 | | gte-Qwen2-1.5B | | NV-Embed-v2 | | gemini-emb | | Average | | Oracle | |
|---|---|---|---|---|---|---|---|---|---|---|---|---|
| MSRS-STORY | R-1 | BScr | R-1 | BScr | R-1 | BScr | R-1 | BScr | R-1 | BScr | R-1 | BScr |
| | R-2 | G-EVAL | R-2 | G-EVAL | R-2 | G-EVAL | R-2 | G-EVAL | R-2 | G-EVAL | R-2 | G-EVAL |
| Llama 2 - 7B | 33.45 | 84.42 | 36.24 | 84.77 | 35.72 | 84.85 | 35.89 | 84.88 | 35.33 | 84.73 | 37.08 | 85.21 |
| | 7.17 | 31.13 | 7.98 | 36.38 | 8.02 | 37.58 | 8.17 | 39.40 | 7.84 | 36.12 | 8.97 | 45.13 |
| Llama 2 - 70B | 34.26 | 84.60 | 36.44 | 85.05 | 36.35 | 85.06 | 36.56 | 85.17 | 35.90 | 84.97 | 37.99 | 85.41 |
| | 7.53 | 32.66 | 8.39 | 39.97 | 8.71 | 40.58 | 8.97 | 42.24 | 8.40 | 38.86 | 9.47 | 48.95 |
| Llama 3.1 - 8B | 34.38 | 84.15 | 35.98 | 84.76 | 36.70 | 84.90 | 37.24 | 84.94 | 36.08 | 84.69 | 37.73 | 85.34 |
| | 7.42 | 37.01 | 8.07 | 44.85 | 8.35 | 46.25 | 8.72 | 48.79 | 8.14 | 44.23 | 9.19 | 55.77 |
| Qwen2.5 - 7B | 33.91 | 84.05 | 36.03 | 84.66 | 36.36 | 84.71 | 36.39 | 84.68 | 35.67 | 84.52 | 38.37 | 85.36 |
| | 6.86 | 38.54 | 7.75 | 46.56 | 8.13 | 48.08 | 8.04 | 50.67 | 7.70 | 45.96 | 9.07 | 57.40 |
| Llama 3.3 - 70B | 34.90 | 84.13 | 36.77 | 84.55 | 36.23 | 84.48 | 37.19 | 84.76 | 36.27 | 84.48 | 38.64 | 85.05 |
| | 7.52 | 40.88 | 8.45 | 49.02 | 8.20 | 50.27 | 8.95 | 51.22 | 8.28 | 47.85 | 9.80 | 58.88 |
| Llama 3.1 - 70B | 34.40 | **84.95** | 36.89 | **85.51** | 37.29 | **85.56** | 36.92 | 85.52 | 36.38 | **85.38** | 39.25 | **86.02** |
| | 7.99 | 41.20 | 9.13 | 50.41 | 9.29 | 50.89 | 9.22 | 51.29 | 8.91 | 48.45 | 10.49 | 59.85 |
| Qwen2.5 - 72B | 35.88 | 84.19 | 38.18 | 84.83 | 38.31 | 84.89 | 38.82 | 85.05 | 37.80 | 84.74 | 40.77 | 85.47 |
| | 7.74 | 43.95 | 8.71 | 54.78 | 8.87 | 55.56 | 9.06 | 59.20 | 8.60 | 53.37 | 10.45 | 67.70 |
| GPT-4o mini | 36.11 | 84.26 | 37.71 | 84.67 | 38.27 | 84.72 | 38.60 | 84.86 | 37.67 | 84.63 | 40.00 | 85.23 |
| | 7.17 | 44.58 | 8.04 | 51.75 | 8.08 | 54.14 | 8.44 | 56.67 | 7.93 | 51.78 | 9.19 | 61.08 |
| GPT-4o | **37.21** | 84.45 | 39.44 | 84.99 | 39.67 | 85.06 | 40.21 | 85.20 | 39.13 | 84.92 | 41.95 | 85.57 |
| | 7.92 | 45.86 | 9.20 | 55.88 | 9.14 | 57.48 | 9.46 | 59.88 | 8.93 | 54.77 | 10.70 | 68.22 |
| Gemini 1.5 Pro | 34.25 | 84.27 | 38.41 | 85.14 | 38.95 | 85.21 | 39.72 | 85.43 | 37.83 | 85.01 | 41.84 | 85.82 |
| | 7.21 | 40.92 | 8.88 | 55.38 | 9.01 | 55.71 | 9.83 | 59.53 | 8.73 | 52.89 | 10.79 | 68.69 |
| Gemini 2.0 Flash | 36.39 | 84.56 | **40.23** | 85.35 | **40.46** | 85.38 | **40.90** | **85.60** | **39.50** | 85.22 | **42.82** | **86.02** |
| | 8.07 | 42.90 | **9.87** | 56.45 | **9.81** | 57.01 | **10.47** | 59.54 | **9.55** | 53.97 | **11.64** | 69.22 |
| DeepSeek-V3 | 37.07 | 84.86 | 39.84 | 85.35 | 39.72 | 85.34 | 40.20 | 85.48 | 39.21 | 85.26 | 42.08 | 85.96 |
| | **8.29** | **46.69** | 9.39 | **57.72** | 9.49 | **58.49** | 10.00 | **61.63** | 9.29 | **56.13** | 10.85 | **69.85** |
| Average | 35.18 | 84.41 | 37.68 | 84.97 | 37.84 | 85.01 | 38.22 | 85.54 | 37.23 | 84.88 | 39.88 | 85.54 |
| | 7.57 | 40.53 | 8.66 | 49.93 | 8.76 | 51.00 | 9.11 | 53.34 | 8.53 | 48.70 | 10.05 | 60.90 |

Table 13: Final generation performance on MSRS-STORY. R-1, R-2, and BScr refer to ROUGE-1, ROUGE-2, and BERTScore F1 scores, respectively. The performance with different retrievers and the oracle document set is reported.

## C Prompt Templates

### C.1 Summarization Prompt Templates

- **MSRS-STORY**:
  *You are a helpful assistant.* [For oracle only] *After reading a lengthy story, provide a 200 to 300 word answer to a question posed about the story.* [For all other retrieval settings] *After reading chapters from various stories, provide a 200 to 300 word answer to a question posed about a specific story.* *Directly respond to the question and do not chat with the user. Answer the following question posed about a story. QUESTION: {query} STORY: {story}*
  Here *{story}* represents a selection of the most relevant story chapters retrieved by a retrieval model based on the query from MSRS-STORY.

- **MSRS-MEET**:
  *You are a helpful assistant. After reading a set of lengthy meeting transcripts, provide a 200 to 300 word answer to a question posed about the meetings. Directly respond to the question and do not chat with the user. Answer the following question posed about the meetings. QUESTION: {query} MEETINGS: {meetings}*
  In this case, *{meetings}* refers to a selection of the most relevant meetings retrieved by a retrieval model based on the query from MSRS-MEET.

| | MSRS-MEET | | | | | | | | | | | |
|---|---|---|---|---|---|---|---|---|---|---|---|---|
| | **BM25** | | **BM25 Rerank** | | **NV-Embed-v2** | | **NV2 Rerank** | | **Average** | | **Oracle** | |
| | R-1 / R-2 | BScr / G-EVAL | R-1 / R-2 | BScr / G-EVAL | R-1 / R-2 | BScr / G-EVAL | R-1 / R-2 | BScr / G-EVAL | R-1 / R-2 | BScr / G-EVAL | R-1 / R-2 | BScr / G-EVAL |
| Llama 2 - 7B | 35.27 | 84.76 | 36.08 | 84.90 | 36.09 | 84.96 | 37.00 | 85.19 | 36.11 | 84.95 | 37.16 | 85.25 |
| | 7.60 | 36.03 | 8.18 | 36.15 | 8.27 | 37.84 | 8.45 | 38.40 | 8.12 | 37.11 | 8.87 | 41.12 |
| Llama 2 - 70B | 36.83 | 85.04 | 37.76 | 85.32 | 37.48 | 85.27 | 36.82 | 85.24 | 37.22 | 85.22 | 38.54 | 85.48 |
| | 8.70 | 37.15 | 9.04 | 39.68 | 8.81 | 40.30 | 8.82 | 39.89 | 8.84 | 39.25 | 9.75 | 42.21 |
| Llama 3.1 - 8B | 28.26 | 83.25 | 30.09 | 83.80 | 28.87 | 83.67 | 29.66 | 83.75 | 29.22 | 83.62 | 29.63 | 83.75 |
| | 6.88 | 37.20 | 7.70 | 41.42 | 6.86 | 39.67 | 7.25 | 41.22 | 7.17 | 39.88 | 7.62 | 44.29 |
| Llama 3.1 - 70B | 34.67 | 84.78 | 35.12 | 84.94 | 35.73 | 85.15 | 35.77 | 85.20 | 35.32 | 85.02 | 37.29 | 85.45 |
| | 8.18 | 39.53 | 8.35 | 41.25 | 8.54 | 40.74 | 8.74 | 43.14 | 8.45 | 41.17 | 9.52 | 45.46 |
| Llama 3.3 - 70B | 32.43 | 82.91 | 34.09 | 83.70 | 34.68 | 83.79 | 34.19 | 83.70 | 33.85 | 83.53 | 35.96 | 84.33 |
| | 7.16 | 40.04 | 7.89 | 43.55 | 8.13 | 43.04 | 7.93 | 43.06 | 7.78 | 42.42 | 8.80 | 47.11 |
| Qwen2.5 - 7B | 35.54 | 84.55 | 36.38 | 84.89 | 36.69 | 84.90 | 36.48 | 84.83 | 36.27 | 84.97 | 37.51 | 85.22 |
| | 7.93 | 39.29 | 7.81 | 43.19 | 8.12 | 41.53 | 8.14 | 43.70 | 8.00 | 41.93 | 8.83 | 48.87 |
| Qwen2.5 - 72B | 34.47 | 84.03 | 35.61 | 84.57 | 35.80 | 84.55 | 35.79 | 84.57 | 35.42 | 84.43 | 37.65 | 84.94 |
| | 8.71 | 43.21 | 9.20 | 46.01 | 8.88 | 44.26 | 9.17 | 46.47 | 8.99 | 44.99 | 10.38 | 53.25 |
| Gemini 1.5 Pro | 36.26 | 84.95 | 37.98 | 85.17 | 37.20 | 85.16 | 38.22 | 85.15 | 37.41 | 85.11 | 39.88 | 85.77 |
| | 7.89 | 37.54 | 8.03 | 40.86 | 8.01 | 39.98 | 8.10 | 40.98 | 8.01 | 39.84 | 9.58 | 47.97 |
| Gemini 2.0 Flash | 37.80 | 85.49 | 38.75 | 85.73 | 38.70 | 85.62 | 38.74 | **85.79** | 38.50 | 85.66 | 41.02 | **86.20** |
| | 8.99 | 40.93 | 9.10 | 43.59 | 8.77 | 41.77 | 9.35 | 45.13 | 9.05 | 42.86 | **10.71** | 52.31 |
| DeepSeek-V3 | 38.48 | **85.54** | 39.49 | 85.72 | **39.31** | **85.71** | 39.10 | 85.69 | 39.09 | 85.66 | 41.16 | 86.13 |
| | **9.04** | 44.23 | **9.44** | 45.95 | **9.25** | 44.97 | 8.97 | 46.53 | 9.17 | 45.42 | 10.29 | 53.52 |
| GPT-4o mini | **38.89** | 85.53 | **39.88** | **85.76** | 39.19 | 85.63 | 39.76 | 85.75 | **39.43** | **85.67** | **41.31** | 86.11 |
| | 8.71 | 45.29 | 9.06 | 47.13 | 8.90 | 45.66 | 8.97 | 47.07 | 8.91 | 46.29 | 9.96 | 53.56 |
| GPT-4o | 38.49 | 85.29 | 39.86 | 85.57 | 39.23 | 85.51 | **39.82** | 85.58 | 39.35 | 85.49 | 41.08 | 86.02 |
| | 8.72 | 45.08 | 9.22 | 46.57 | 9.19 | 45.73 | 9.58 | 46.85 | **9.18** | 46.06 | 10.64 | 53.67 |
| Average | 35.62 | 84.68 | 36.76 | 85.01 | 36.58 | 84.99 | 36.78 | 85.04 | 36.43 | 84.93 | 38.18 | 85.39 |
| | 8.21 | 40.46 | 8.58 | 42.95 | 8.48 | 42.12 | 8.62 | 43.54 | 8.47 | 42.27 | 9.58 | 48.61 |

Table 14: Final generation performance on MSRS-MEET.

## C.2 Query Rewriting Prompt Templates:

- **MSRS-STORY:**
  *You are given a question about a story and the full story itself, which is divided into chapters separated by <doc-sep>. Your task is to rewrite the question so that it includes a brief, vague description of the story while preserving the structure and all of the details of the original question. However, do not introduce specific details from the story that would reveal the answer. Ensure that the resulting question remains a single sentence. STORY: {story} QUESTION: {query}*

## C.3 Prompt Templates for Scoring Gold Documents by Relevance:

- **MSRS-STORY:**
  *You are given a question about a story and the full story itself, which is divided into chapters separated by <doc-sep>. You are also given a specific chapter from the story. Your task is to assign a score from 1 to 10 on how relevant this specific chapter is to answering the question compared to the other chapters in the story. Respond with "SCORE: " followed by your score. On the next line, respond with "EXPLANATION: " followed by a brief two-sentence explanation of your rationale for choosing the score. QUESTION: {query} STORY: {story} CHAPTER: {chapter}*

- **MSRS-MEET:**
  *You are given a question about a discussion that happened over multiple meetings and a specific meeting transcript. Your task is to assign a score from 1 to 10 on how relevant this specific meeting transcript is to answering the question. Respond with "SCORE: " followed*

*by your score. On the next line, respond with "EXPLANATION: " followed by a brief two-sentence explanation of your rationale for choosing the score.*

## C.4 Prompt Templates for Reranking Retrieved Documents Pointwise:

- **MSRS-STORY:**
  *You are given a question about an unknown story and a chapter from a specific story. Your task is to assign a score from 1 to 20 based on how relevant this chapter is to answering the question. Respond only with a numerical score. QUESTION: {query} CHAPTER: {chapter}*

- **MSRS-MEET:**
  *You are given a question about a discussion that happened over multiple meetings. You are also given the transcript of a meeting. Your task is to assign a score from 1 to 20 based on how relevant this specific meeting transcript is to answering the question. Respond with only a numerical score. QUESTION: {query} MEETING: {meeting}*

## C.5 G-EVAL Prompt Templates

- **G-EVAL consistence:**
  *You will be given one prediction article and one reference article. Your task is to rate the prediction on one metric. Please make sure you read and understand these instructions carefully. Please keep this document open while reviewing, and refer to it as needed. Evaluation Criteria: Consistency (1-5) - the factual alignment between the prediction and the reference. A factually consistent prediction contains only statements that are entailed by the reference.*
  *Evaluation Steps: 1. Read the reference carefully and identify the main facts and details it presents. 2. Read the prediction and compare it to the reference. Check if the prediction contains any factual errors that are not supported by the reference. 3. Assign a score for consistency based on the Evaluation Criteria.*

- **G-EVAL relevance:**
  *You will be given one prediction article and one reference article. Your task is to rate the prediction on one metric. Please make sure you read and understand these instructions carefully. Please keep this document open while reviewing, and refer to it as needed. Evaluation Criteria: Relevance (1-5) - selection of important content from the source. The prediction should include only important information from the reference.*
  *Evaluation Steps: 1. Read the prediction and the reference carefully. 2. Compare the prediction to the reference and identify the main points of the reference. 3. Assess how well the prediction covers the main points of the reference. 4. Assign a relevance score from 1 to 5.*

## C.6 Data Contamination Prompt Templates

- **MSRS-STORY:**
  *You are a helpful assistant who has seen the provided story from the SQuALITY dataset before. Provide a 200 to 300 word answer to a question posed about the story. Rely upon your internal knowledge of the story and give it your best guess. Answer the following question posed about the story {title} by {author}. QUESTION: {query}*

- **MSRS-MEET:**
  *You are a helpful assistant who has seen the meeting transcripts from the QMSum dataset before. Provide a 200 to 300 word answer to a question posed about these meetings. Rely upon your internal knowledge of the meetings and give it your best guess. Answer the following question posed about the meetings. QUESTION: {query}*

# D   Models

## D.1   Retrieval Details

Retrieval settings were created using the top $K$ retrieved documents. We chose $K = 8$ for MSRS-STORY and $K = 3$ for MSRS-MEET based on the average number of oracle documents per query (8.79 and 3.04, respectively).

We observed that dense retrievers performed best when chunks with a maximum size of 1000 tokens were embedded, and the chunk embeddings were averaged (weighted by chunk size) to produce the overall embedding. Therefore, we applied this approach to all the dense retrieval models we tested to effectively handle documents of all lengths.

## D.2   Inference Details

Experiments were primarily conducted on Nvidia H100 GPUs, along with A100 and A6000 GPUs. We loaded all open-source models with 4-bit `bitsandbytes` quantization and performed inference using the vLLM framework (Kwon et al., 2023). For all non-reasoning models, we used a temperature of 0.7, a top-p value of 0.9, and a maximum output token limit of 600 to accomodate the 95th percentile reference summary lengths (601.00 and 323.75 tokens for MSRS-STORY and MSRS-MEET, respectively). For reasoning models, we used the default reasoning effort level: "medium" for OpenAI models and "dynamic thinking" for Gemini models.

For the Llama 2 models (Touvron et al., 2023) with a maximum context length of 4096 tokens, the input documents were truncated to fit within this window. For the rest of the models, no truncation of input documents was used for MSRS-STORY.

For MSRS-MEET, the input documents were truncated to fit within a 32k-token context window for all models in Table 7, ensuring that locally hosted models fit on the GPUs and that configurations were consistent across models. This context limit comfortably accommodates the three retrieved documents in MSRS-MEET, which average roughly 22k tokens in total (Table 2). Furthermore, we tested summarization without truncation for a few models in the oracle setting of MSRS-MEET, and found that the evaluation results were very similar. No truncation was applied to the reasoning models in Table 8.

## D.3   Evaluation Models

### D.3.1   Retrieval-Evaluation Models

Normalized Discounted Cumulative Gain (NDCG) (Järvelin & Kekäläinen, 2000) evaluates the quality of a ranked list by assigning greater weight to relevant documents that appear earlier in the ranking. Mean Average Precision (MAP) (Voorhees & Harman, 1999) calculates the mean of the precision scores at the ranks where relevant documents are retrieved, reflecting both precision and recall across the entire ranking. Precision@$K$ (P@$K$) is the proportion of relevant documents among the top $K$ retrieved, while Recall@$K$ (R@$K$) measures the proportion of all relevant documents that are included in the top $K$. We use $K = 8$ for MSRS-STORY and $K = 3$ for MSRS-MEET, aligning with the retrieval setups detailed in Appendix §D.1.

### D.3.2   Summarization-Evaluation Models

**ROUGE** (Lin, 2004) is a classic metric for summarization evaluation, measuring the word overlap between the candidate and reference summaries. We compute all ROUGE metrics against the four reference summaries in MSRS-STORY and the single summary in MSRS-MEET, using F1 scores for evaluation.

**BERTScore** (Zhang et al., 2020) calculates the similarity between the reference and generated summary using contextual word embeddings. We use F1 scores to evaluate BERTScore.

**G-EVAL** (Liu et al., 2023) is an LLM-as-a-judge framework that employs LLMs with chain-of-thought (CoT) prompting and rubrics to assess the quality of outputs. Given the original text, a question, and a generated answer, an LLM judge is instructed to score the answer across various dimensions (e.g., consistency, coherence, relevance, fluency). G-EVAL is able to achieve better performance than ROUGE and BERTScore across different dimensions of summarization evaluation on the SummEval (Fabbri et al., 2021) benchmark.

**G-EVAL Implementation Details**   We used GPT-4o (model version: `gpt-4o-2024-08-06`) as the backbone of G-EVAL. Here we compare the consistency and relevance between the predicted summary and the reference summary. Consistency measures whether the information in the predicted summary is faithful to the reference summary, while relevance measures the semantic similarity between them. For evaluation, we report G-EVAL scores computed using the relevance rubric (Appendix §C.5) as it is the only G-EVAL rubric that directly assesses semantic similarity between texts.

We note that different from the original implementation of G-EVAL, we use G-EVAL for *reference-based* evaluation as the input documents can become prohibitively long. Besides, it enables a fair comparison with ROUGE and BERTScore, which are reference-based. Instead of using the model's direct output, we compute an average score based on the model's log probabilities. This follows the approach described in G-EVAL (Liu et al., 2023), which allows for more nuanced, continuous scoring that better reflects the quality of the generated texts. Specifically, the normalized probabilities of the tokens "1", "2", "3", "4", and "5" are weighted by the numerical values of these tokens to calculate a weighted average score between 1 and 5 (prompts included in Appendix §C.5).

# E   Human Validation Criteria and Guidelines

*Summary Coherence and Fluency*
Measures the overall language quality and coherence of the summary.

- **5:** The summary is coherent, well-organized, and easy to read.

- **3:** The summary is mostly understandable but has some issues with coherence and clarity. The flow may be somewhat disjointed and the language a bit difficult in places.

- **1:** The summary is very difficult to understand due to severe issues with coherence and organization.

*Factual Consistency and Accuracy*
Assesses the factual alignment between the summary and the retrieved source documents.

- **5:** The summary is completely factually consistent with the information provided in the retrieved documents.

- **3:** The summary is mostly factually consistent with the source documents, but contains some minor inconsistencies.

- **1:** The summary has severe factual inconsistencies with the source documents, including inaccuracies, contradictions or unsupported (hallucinated) claims.

*Open-Domain Retrieval Necessity*
Assesses the extent to which the query requires retrieving information from a large corpus ($\geq 3$), as opposed to a narrow set ($<3$) of pre-selected documents.

- **5:** The query requires searching a large open-domain corpus to find relevant information.

- **3:** The query could potentially be answered using a focused set of documents, but searching a larger open-domain corpus provides additional relevant information.

- **1:** The query could be fully answered using a small pre-defined set of documents.

*Multi-Document Information Aggregation*
Measures how well the summary synthesizes and aggregates information from multiple retrieved documents to answer the query. This also reflects the effectiveness of merging summaries by the LLM used in our study.

- **5:** The summary effectively aggregates and synthesizes information from multiple retrieved documents into a coherent answer to the query.
- **3:** The summary uses some information from multiple documents but mainly relies on one or two dominant documents to answer the query.
- **1:** The summary fails to aggregate information from multiple documents at all.

*Query Relevance and Coverage*
Evaluates how well the summary captures the key information requested in the query.

- **5:** The summary directly and comprehensively addresses all parts of the query, covering the most relevant information.
- **3:** The summary partially addresses the query, covering some relevant information but missing other important aspects.
- **1:** The summary does not address the query or capture any of the key relevant information.

*Inter-Document Relationship Capture*
Evaluates how well the summary captures and expresses the relationships between information across different retrieved documents.

- **5:** The summary well reflects the relationships between relevant information from different documents.
- **3:** The summary captures some relationships between information from different documents.
- **1:** The summary does not capture or express any relationships between information from different documents.

### E.1 Annotation Guideline

1. Read the query carefully to understand the information needed.
2. Read through the documents to assess their relevance to the query.
3. Read the summary and proceed to evaluate the entire example on the 6 criteria above, assigning a score from 1-5 for each criterion.
4. For each score, provide a brief justification noting key observations and examples from the summary and documents.
5. Add further comments if there is any.

## F Dataset Example

We show one example of each subset of the MSRS dataset in Table 15 and Table 16.

| Field | Content |
|---|---|
| *Query* | In a story where two men navigate a tense and dangerous situation involving a power source that could revive a struggling planet, describe the setting of the story. |
| *SQuALITY Query* | Describe the setting of the story. |
| *Gold Document #1* | ...Ryd Randl stood, slouching a little, in the darkened footway, and watched the sky over Dynamopolis come alive with searchlights. The shuttered glow of Burshis' Stumble Inn was only a few yards off to his right, but even that lodestone failed before the novel interest of a ship about to ground in the one-time Port of Ten Thousand Ships. Now he made out the flicker of the braking drive a mile or so overhead, and presently soft motor thunder came down to blanket the almost lightless city with sound. A beam swayed through the throbbing darkness, caught the descending ship and held it, a small gleaming minnow slipping through the dark heavens. A faint glow rose from Pi Mesa, where the spaceport lay above the city, as a runway lighted up—draining the last reserves of the city's stored power, but draining them gladly now that, in those autumn days of the historic year 819, relief was in sight... |
| *Gold Document #2* | ...And that was it. The almost airless Martian sky, with its burning actinic rays, is so favorable for the use of the helio-dynamic engine. And after the middle of the eighth century, robot labor gave Mars its full economic independence—and domination. For power is—power; and there is the Restriction Act to keep men on Earth even if more than two in ten could live healthily on the outer world. "Ten years ago," Mury nodded as if satisfied. "That must have been the Power Company of North America—the main plant by Dynamopolis itself, that shut down in December, 809. They were the last to close down outside the military bases in the Kun Lun." Ryd was pacing beside him now. He felt a queer upsurge of confidence in this strange man; for too long he had met no sympathy and all too few men who talked his language. He burst out: "They wouldn't take me, damn them! Said my record wasn't good enough for them. That is, I didn't have a drag with any of the Poligerents"... |
| *Gold Document #3* | ...They emerged in shadow, hugging the wall. Almost a quarter of a mile to the right the megalith of the Communications Tower, crowned with many lights where the signal-men sat godlike in its summit. Its floodlights shed a vast oval of light out over the mesa, where the mile-long runways—no longer polished mirror-like as in the days of Dynamopolis' glory—stretched away into the darkness of the table land. A handful of odd ships—mere remnant of the hundreds that Pi Mesa port had berthed—huddled under the solenoid wickets, as if driven together by the chill of the thin, knife-like wind that blew across the mesa... |
| *Gold Summary #1* | The story takes place in Dynamopolis, a city in North America, in the year 819. The city is flooded with searchlights, although there is very little power to go around. The Terrestrials must gather at the local bar, Stumble Inn, if they do not want to freeze to death. At one point, Dynamopolis was a wealthy city, known as the Port of Ten Thousand Ships. About ten years ago, the Power Company of North America and the Triplanet Freighting Company were shut down, and the majority of the Terrestrials lost their jobs. The only people with political power are the Poligerents, and unless a Terrestrial knows one of them, he or she is likely left without a way to make ends meet. The Terrestrials were recently told that the power will be restored once the power shell is put on Earth. The air is thin, but the Terrestrials have become accustomed to it. Pi Mesa is the spaceport that hovers over the city. There are still unused ships hovering there from the days where it was an important port with lots of action. Just outside of Pi Mesa there are hundreds of low buildings that are abandoned because they are no longer useful. They contain fuel pumps and servicing equipment, and they serve as a constant reminder of the life the Terrestrials once lived. When Ryd and Mury break into the land patrolled by the guards in blue in the spaceport, they find narrow passages, spiral staircases, and cool metal walls covered in dust. The Communications Tower is nearby, and it is guarded by signal-men. The soldier robots that are on patrol are about as tall as the average Terrestrial, and they are scarlet colored. They are unarmed and are mostly there to scare intruders away. Mury and Ryd aim to get on a ship called Shahrazad, which rests on the Number Two Runway, waiting for takeoff. When they enter the ship, they find that the cabin is very hot and full of dials and needles. There is a curved control panel in front, and the ship makes a humming sound because of all of the air-purifiers onboard. Mars is an important setting in the story, although the characters do not actually travel there. Mars is almost airless, so it is very easy to run a helio-dynamic engine. On Mars, they use robots for labor, and due to a law that has been passed, Terrestrials are forced to stay on Earth. |
| *Gold Summary #2* | ... |

Table 15: Example from the MSRS-STORY Dataset

| Field | Content |
|---|---|
| *Query* | Summarize the group's discussion on the function and object of the remote control, including the kinetic function and the ability to roll through the user's favorite channels. |
| *Gold Document #1* | ...marketing: because you have you can implement a lot of buttons in one remote with not that much buttons . Especially the volume buttons , the channel buttes buttons and the number buttons to zap through the channels . project manager: then I think it's a good thing that I made a separate slide of them marketing: Ja , project manager: so you can all read them . project manager: So that's the first thing we I think we should pay less attention to teletext . the remote control should only be used for the television , otherwise the project becomes more complex , which endangers the time to market , and of course would make it more costly , I think . our current customers are within the age group of forty plus , and new product should reach a new market with customers that are younger than forty, and you talked about that before . And a last point , but also very important , our corporate image should stay recognisable in our products , which means that our corporate colour and slogan must be implemented in the new design... |
| *Gold Document #2* | ...user interface: Well , what about this what about if you can programme in your favourite channels into this scroll wheel and you can just like roll through your favourite channels , project manager: . user interface: and it c it project manager: You'd need a display on the th the thing . user interface: Why ? It'll tell you when you flip the channel on the T_V_ . You say programme start , and then type in marketing: Put user interface: 'cause you still have the typing you know you'll still have the keypad where you can type 'em in manually . user interface: So programme start , zero , one , enter , zero , five , enter , thirty eight , enter , programme end . project manager: and that just basically flips between it and it'll go it sends out zero , five , and then thirty six , and then zero , one again . project manager: That's not gonna be too expensive because that's gonna be you're gonna be able to nab that off of computer mouse manufacturers really . user interface: Oh well we also have to determine in some manner how to switch between modes , between going through your favourites list and just hitting up one , up two . marketing: Or we go directional up we go we go this we go this we go this way for one , we go this way for the other . project manager: Yeah people are gonna have their favourite sorta , whether they do that or whether they marketing: Ah-ha okay . user interface: I think we'll need a we'll need a mode switch , but then if we have a mode switch we're gonna need some kinda indicator... |
| *Gold Document #3* | ...industrial designer: Yeah , we just make it flat . industrial designer: But , you do l marketing: But , wha 'Kay , look , what is the If you make it double-curved , it costs one Euro more . marketing: fun function more like industrial designer: Worth , does it have added worth ? user interface: there's an a a athe aesthetic value , but not functionality . industrial designer: R if you promote a kinetic I kinetic remote control , that would b sell better than an a normal remote control . industrial designer: No , well , y , y you can go into your neighbour and tell him , ha , my k remote control is kinetic . marketing: What a what about all the m the environment freaks ? user interface: Yeah , but it doesn't fit in our co cost profile . project manager: Yeah ? Who because if you want to go to kinetic , you're you're on thirteen and a half and you must go to flat... |
| *Gold Summary* | The user interface designer suggested a function where users can start a program and input their favorite channels by pressing a button, which was well-received by the project manager and marketing team. The cost of implementing this function was deemed manageable through potential partnerships or utilizing existing technology. Additionally, a mode switch and indicator were deemed necessary for this function. The marketing team proposed designing a remote control solely for television, while the user interface designer suggested including buttons for video recorder functions, although unsure of the technological feasibility. The industrial designer agreed that a simpler remote control for television would benefit older or less coordinated individuals. The initial design of the remote control was discussed, with an emphasis on user-friendliness and clear buttons. The project manager suggested paying less attention to outdated features like teletext and focusing on customers aged forty and above, incorporating the corporate color and slogan. The remote control was intended to work exclusively with TVs, with basic functions such as volume, channel, and number buttons. The user interface proposed a button that can switch between one and two numbers. The idea of a stand-alone remote control with customizable faces and functionality for other devices was suggested. There was a disagreement between the project manager and industrial designer regarding the inclusion of kinetic function, with the manager considering it a cost reduction measure while the designer believed it would be a desirable marketing feature. To maintain the price level, they decided to adapt the control into a flat design. |

Table 16: Example from the MSRS-MEET Dataset

