# OpenReview forum: "MSRS: Evaluating Multi-Source Retrieval-Augmented Generation"
_colmweb.org/COLM/2025/Conference — COLM 2025_

### Official Review · Reviewer_JeWr · 2025-04-19

**Rating:** 6
**Confidence:** 3
**Ethics Flag:** 1

**Summary:**

This paper presents two novel datasets for open-domain, query-based multi-document summarization, and an extensive evaluation of retrievers and generative models. The work is designed as an improvement over Giorgi et al. EMNLP 2023 in terms of both query quality and document corpus.

The two datasets have been obtained from two corresponding datasets for long-context, single-document, question-based summarization through a process of subdivision and re-annotation.

Ample results for the retrievers and the generators are presented in Sections 5.1 and 5.2, respectively.

**Questions To Authors:**

My questions above are about the construction and annotation of the datasets, which is the key stated contribution.

***

After the discussion phase, I have raised my rating.

**Reasons To Accept:**

1.	The evaluation of retrievers and generators is comprehensive and contemporary.

2.	The paper is well written and easy to read.

3.	All the code and data have been generously shared in a public repository.

**Reasons To Reject:**

1.	I have various reservations about the validity of the benchmarks for the designated task. For OdSum-Story, the original long documents have been segmented into shorter documents using natural HTML delimiters such as hr: have these been used consistently by the authors? What is the variance over the length? And, more critically, who has decided what are the ground-truth documents for a specific query? If I understand correctly, this is the key annotation for the performance evaluation of the retrievers.

2.	The description of the construction of OdSum-Meet addresses query and summaries, but not the segmentation into documents and the annotation of the ground-truth set of documents.

3.	Also the human evaluation of these datasets does not seem to have assessed the correctness of the ground-truth annotation (yes for precision, not for recall). Overall, it looks as if the hardest part of the retrieval annotation has been skipped.

4.	On a separate note, I believe the title is misleading: the paper does not cover “generation” at large, but only query-based summarization, and should be retitled as such.

---

> ### Author Response · Authors · 2025-06-03
> **Rebuttal**
>
> Thank you for reviewing our work! We would like to address your concerns and questions as follows:
>
> > For OdSum-Story, the original long documents have been segmented into shorter documents using natural HTML delimiters such as hr: have these been used consistently by the authors? What is the variance over the length?
>
> We segmented the original story texts from SQuALITY into the chapter documents of ODSum-Story using HTML delimeters. Specifically, we applied the following procedure uniformly across the dataset: split stories by `<hr class="chap"/>`, strip leading white-space characters and HTML tags, and collect the non-empty segments as chapters. To verify the consistency and quality of our segmentation process, we report the following statistics for ODSum-Story.
> * Number of stories: 127
> * Number of documents (i.e. chapters): 1,138
> * Average number of chapters per story: 8.96
> * Average document length: 556.93 words / 845.44 tokens
> * Global standard deviation of document lengths: 382.07 words / 577.29 tokens
> * Average per-story standard deviation of document lengths: 307.28 words / 465.31 tokens
>
> As shown in the provided histogram (anonymous link: https://postimg.cc/WF9vRhMQ), document lengths are tightly clustered around the mean, indicating that most segments are of comparable length. These statistics highlight the natural distribution of chapter lengths in SQUaLITY, which our segmentation process preserves in ODSum-Story.
>
> > And, more critically, who has decided what are the ground-truth documents for a specific query? If I understand correctly, this is the key annotation for the performance evaluation of the retrievers.
>
> We fully agree that the ground-truth annotation is central to the evaluation of retrieval performance, and we discuss this point further in a later response. Regarding ODSum-Story, each query is derived from SQuALITY, where queries are answerable using chapters within a single story. Accordingly, we define the ground-truth as the full set of chapters from the sole relevant story.
>
> > The description of the construction of OdSum-Meet addresses query and summaries, but not the segmentation into documents and the annotation of the ground-truth set of documents.
>
> For ODSum-Meet, the segmentation and ground-truth annotation were derived from the query clustering process described in Appendix B. Each QMSum query is associated with an annotated set of relevant meeting transcripts, which we use without modification as the documents in ODSum-Meet. We cluster similar QMSum queries based on embedding similarity to form a broader, merged query. Our human verification in Table 3 confirms the high quality of the queries and summaries formed through this process. The ground-truth set for each merged query is defined as the union of the ground-truth documents for the constituent queries.
>
> > Also the human evaluation of these datasets does not seem to have assessed the correctness of the ground-truth annotation (yes for precision, not for recall). Overall, it looks as if the hardest part of the retrieval annotation has been skipped.
>
> For ODSum-Story, each query corresponds to a single story, and we include all chapters of that story in the ground-truth set, ensuring that the ground-truth annotations are accurate in both precision and recall. For ODSum-Meet, we acknowledge that additional relevant documents may exist beyond the ground-truth set, perhaps due to overlaps in meeting content that may not have been captured by the original QMSum annotations or our query clustering process.
>
> To account for this, we report G-Eval scores, which assess retrieval quality through downstream summarization performance, alongside standard retrieval metrics in Tables 4 and 5. Notably, G-Eval reflects the contribution of all retrieved documents, not just those in the annotated ground-truth set. As a result, even if a relevant retrieved document is not in the ground-truth, it can still boost the G-Eval score by providing helpful context for a better summary.
>
> > On a separate note, I believe the title is misleading: the paper does not cover “generation” at large, but only query-based summarization, and should be retitled as such.
>
> Thank you for this suggestion. While our task is centered around query-based summarization, we use the term "retrieval-augmented generation" in the title to reflect the importance of both retrieval and generation in the benchmarks we propose. The central challenge in our benchmarks lies in generating natural language outputs that involve not only summarization based on a user query, but also filtering and integrating relevant content from multiple “retrieved” documents.
>
> That said, we recognize the reviewer’s concern that the current title may suggest a broader coverage of generation tasks beyond query-based summarization. We will revise the paper to clarify the generative nature of the task, and will revisit the framing of the title and abstract to ensure that the scope is accurately conveyed.

---

> > ### Comment · Reviewer_JeWr · 2025-06-10
> >
> > I thank the authors for their very clear and patient explanations. Just to politely reply, my concerns about the ground truth for the retrieval task remain: for OdSum-Story, the assumption that the all and only relevant chapters are those of the original document -- some of the chapters in the document may not be relevant to the query at all, and other relevant chapters may be in other documents; for OdSum-Meet, that the union set of the ground truth of the constituent queries can be regarded as the ground truth for the merged query.
> > I acknowledge that all the other reviewers have had much more positive ratings than my own (6, 7, 8 currently) and I am glad to increase mine, because the paper is really well executed and documented.

---

### Official Review · Reviewer_QKAz · 2025-05-11

**Rating:** 8
**Confidence:** 4
**Ethics Flag:** 1

**Summary:**

The paper presents a new benchmark for open-domain multi document summarisation (ODMDS). The authors leverage two pre-existing single document query oriented long-context summarisation datasets in the stories (SQuALITY) and meetings domains (QMSum). They convert the stories dataset into a ODMDS dataset by breaking the stories into chapters (each chapter is considered a document) and decontextualizing the query/question by re-writing it with an LLM. A question may seek information that is present in multiple chapters. For the meetings dataset they cluster queries and summaries in domains, merging similar queries and summaries into a single more comprehensive ones. The generated new resources are human validated for quality. Finally, the author test various RAG+LLM pipelines to assess the performance of various models in the newly generated benchmark. In terms of retrievers, dense retrievers seem to clearly outperform sparse one in the stories dataset, while sparse retrievers are more competitive in the more challenging meetings dataset. In terms of LLMs, DeepSeek v3 performs the best in the stories dataset while GPT-4o (and mini) perform the best in the meets dataset. The LLMs performance vary depending on the retriever.

**Reasons To Accept:**

- The paper is well written and the ideas clearly presented
- Interesting related work section clearly describing the differences of multi-document summarisation (MDS), query focused MDS and Open-domain MDS.
- Valuable resource for the research community to evaluate state-of-the-art ODMDS models
- Well described generation of the dataset. Similar approaches may be used to generate more ODMDS datasets in other domains.

**Reasons To Reject:**

- I would have appreciated more analysis on that justify certain decisions on the creation of the dataset. For example, in the ODSum-Story dataset construction, the authors give a good rational for decontextualizing queries, as the queries in the original dataset are too dependent on the source document. However, it would be interesting to see that this step actually makes a difference in terms of downstream performance. Do the models perform better when these decontextualized queries are provided?

---

> ### Author Response · Authors · 2025-06-03
> **Rebuttal**
>
> Thank you for reviewing our work! We agree that query decontextualization is a critical step in the benchmark creation process that deserves further analysis, so we conducted an additional experiment to directly evaluate its impact on both retrieval and downstream summarization performance.
>
> > Do the models perform better when these decontextualized queries are provided?
>
> Yes, our experiment demonstrates that decontextualized queries lead to consistently better performance across both traditional retrieval metrics (Precision, Recall, NDCG, MAP) and downstream generation quality (G-Eval). This aligns with our motivation for decontextualization: while many original SQuALITY queries rely heavily upon a provided story document for meaning, our benchmark requires queries to stand alone in order to simulate realistic retrieval conditions.
>
> The decontextualization process does not remove any details or modify the intent of the queries, but rather introduces minimal, generic hints that help make the queries retrievable. We include the following example on line 164:
>
> > For instance, a generic query from SQuALITY like "Describe the setting of the story." was transformed into a more context-specific query such as "In a story where two men navigate a tense and dangerous situation involving a power source that could revive a struggling planet, describe the setting of the story."
>
> We first analyzed the original SQuALITY queries from which the 260-query test set for ODSum-Story was derived and found that many lacked sufficient standalone information. For example, 52 queries were slight variations of "What is the plot of the story?"—asking about the plot without providing any specific context. An additional 30 queries asked about the setting in similarly unspecified ways, such as "What is the setting of the story?" or "Describe the setting of the story." In total, nearly one-third of these original SQuALITY queries lacked enough context to support meaningful retrieval without access to the full story.
>
> We evaluated the impact of query decontextualization using both a sparse retriever (BM25) and a strong dense retriever (gemini-embedding), comparing performance between the decontextualized queries in the ODSum-Story test set and their original counterparts from SQuALITY. Both sets of queries map to the same set of gold-standard summaries, isolating the effect of decontextualization on retrieval and downstream summarization performance.
>
> In the tables below, the "Decontextualized Queries" results are from Table 4 in the paper, and the "Original SQuALITY Queries" results were obtained during this experiment. As mentioned in Table 4, G-Eval #1 and G-Eval #2 refer to the G-Eval scores for the summaries produced by Qwen2.5-7B and Gemini 2.0 Flash, respectively, for each retrieval setting.
>
>
> | | Retrieval Setting | P@8 | R@8 | NDCG | MAP | G-Eval #1 | G-Eval #2
> |-|--------------------|------|-------|---------|-------|------------|-------------|
> Decontextualized Queries | BM25 | 20.91 | 23.20 | 28.75 | 18.33 | 38.54 | 42.90
> Original SQuALITY Queries | BM25 |19.04 | 19.16 | 24.49 |16.60 | 33.56 | 36.84
>
>
>
> | | Retrieval Setting | P@8 | R@8 | NDCG | MAP | G-Eval #1 | G-Eval #2
> |-|--------------------|------|-------|---------|-------|------------|-------------|
> Decontextualized Queries|gemini-embedding|56.68 | 57.46  | 72.33 | 54.25 | 50.67 | 59.54
> Original SQuALITY Queries | gemini-embedding | 34.52 | 34.91 | 43.86 | 32.11 | 39.68 | 46.89
>
> The results show a clear and consistent benefit from query decontextualization across all retrieval settings. With BM25, decontextualized queries outperform the original SQuALITY queries on all metrics, including a 5+ point gain in G-Eval scores, demonstrating that even sparse retrievers benefit from more self-contained queries.
>
> The effect is even more pronounced with gemini-embedding, where retrieval metrics improve dramatically (e.g., NDCG increases from 43.86 to 72.33), and G-Eval scores improve by 11+ points. These results confirm that decontextualization enables more effective retrieval by making previously underspecified queries solvable and more suited to the retrieval needs of real-world RAG settings.

---

> > ### Comment · Reviewer_QKAz · 2025-06-07
> >
> > Thanks for the additional analysis. It supports the rationale for decontextualizing the queries. I will raise my score to 8.

---

### Official Review · Reviewer_Witb · 2025-05-13

**Rating:** 7
**Confidence:** 4
**Ethics Flag:** 1

**Summary:**

The paper introduces ODSUM, a set of benchmarks designed to evaluate Retrieval-Augmented Generation (RAG) systems that require multi-document reasoning and synthesis capabilities. Traditional RAG systems typically operate on single-document retrieval and generation, but many real-world applications demand the integration of information from multiple sources. The authors propose a scalable approach for constructing multi-source RAG datasets from query-focused summarization datasets, resulting in two new benchmarks: ODSUM-STORY and ODSUM-MEET, that test retrieving complementary information from large data sources based on given queries. These benchmarks require systems to retrieve and synthesize information from large corpora in response to realistic queries that reflect users' information needs. The paper outlines the construction process, experimental setup, and results, highlighting significant variations in retrieval effectiveness across different tasks and methods.

**Questions To Authors:**

see above

**Reasons To Accept:**

- The proposed datasets that setup real world queries that actually require collecting information across multiple documents is a very valuable resource and thus the introduction of ODSUM-STORY and ODSUM-MEET provides new, challenging benchmarks for evaluating multi-document RAG systems, addressing a gap in existing benchmarks that often use unrealistic queries and datasets.

- The proposed method for constructing multi-source RAG datasets from query-focused summarization datasets is scalable and ensures the integrity of gold summaries while making them applicable in broader open-domain scenarios. This may help inspire other multi-document setups beyond summarization for further research.

- The paper has a rigorous and extensive set experiments on various recent models, RAG system components, including retrieval and generation, providing valuable insights into their capabilities and limitations.

**Reasons To Reject:**

- While there is a fair attempt made to provide some data on the data quality through data annotation, 40 random examples may not be representative of the quality of the entirety of the dataset which is order of magnitudes larger. This also makes the validation of the quality of future datasets generated using this framework not as scalable as they are always limited by human annotation

- It may have also been insightful to see an error analysis of what the models are most challenged in attempting successfully as this may shed light on what the model's current capabilities are able to achieve compared to what is something newer models need to work towards

---

> ### Author Response · Authors · 2025-06-03
> **Rebuttal**
>
> Thank you for reviewing our work! We would like to address your concerns and questions as follows:
>
> > While there is a fair attempt made to provide some data on the data quality through data annotation, 40 random examples may not be representative of the quality of the entirety of the dataset which is order of magnitudes larger. This also makes the validation of the quality of future datasets generated using this framework not as scalable as they are always limited by human annotation
>
> The quality of the multi-document RAG benchmarks created using our methods depends heavily on the quality of the underlying datasets—SQuALITY and QMSum, in our case. The summaries in SQuALITY were written by a team of expert writers, with each query paired with four human-written reference summaries validated for correctness by four reviewers. For each query in QMSum, annotators rigorously marked all relevant text spans, which were then reviewed for accuracy and used as the basis for query-focused summaries generated via LLMs.
>
> Therefore, we expect the results in Table 3 from our dataset quality human evaluation study, which sampled roughly 6% of queries in ODSum-Story and 9% in ODSum-Meet, to scale to larger sample sizes due to the high quality of the underlying datasets. Furthermore, we will verify this claim by reviewing additional samples and updating Table 3 with the results from a larger-scale evaluation in the revised manuscript.
>
> > It may have also been insightful to see an error analysis of what the models are most challenged in attempting successfully as this may shed light on what the model's current capabilities are able to achieve compared to what is something newer models need to work towards
>
> We conducted a small-scale error analysis study with the top-performing models in Tables 6 and 7 for the oracle setting. These models were DeepSeek-V3 for ODSum-Story (G-Eval score of 69.85) and GPT-4o for ODSum-Meet (G-Eval score of 53.67). We plan to conduct a larger-scale error analysis in the revised manuscript.
>
> For this study, forty summaries—twenty per subset—were randomly sampled from the pools of summaries generated by DeepSeek-V3 given the ODSum-Story oracle documents and GPT-4o given the ODSum-Meet oracle documents. The evaluations were divided up equally among two annotators, who selected all the error type labels that applied to each generated summary given the query, the gold-standard summary, and the oracle documents as reference. The table below depicts each error type label along with the prevalence of that error for the two subsets.
>
> Error Type  | Error Prevalence in ODSum-Story Oracle-DeepSeek-V3 Subset (N = 20) | Error Prevalence in ODSum-Meet Oracle-GPT-4o Subset (N = 20)
> -|-|-
> Missing many minor, fine details | 40% | 60%
> Missing many critical, important details | 0% | 35%
> Minor query misunderstanding | 0% | 5%
> Major query misunderstanding | 0% | 10%
> Presence of hallucinations | 10% | 10%
> Excessively general or vague response | 0% | 10%
>
> Our error analysis study indicates that the generated summaries in the subsets often lacked many minor details (40% for ODSum-Story and 60% for ODSum-Meet), with ODSum-Meet also lacking many major details (35% of the time), and additional error types appearing less frequently. These initial findings are consistent with works [1] [2] [3] that demonstrate that even frontier LLMs may still find multi-document tasks difficult due to limitations in multi-document abilities. Furthermore, our human evaluation found ODSum-Meet to be more difficult than ODSum-Story, which aligns with the delta in G-Eval scores across the subsets (53.67 vs. 69.85). This observation is also supported by the original QMSum benchmark paper and recent works [4] [5] performing evaluations upon QMSum that attest to this benchmark’s difficulty.
>
> We refer to the existing results in our paper to help analyze the remaining error beyond the oracle setting. As shown in Tables 6 and 7, the summarization performance of models, even for advanced generators, is heavily dependent on the quality of the retrieved documents. Additionally, Table 8 in the appendix illustrates that even having access to the most relevant oracle document underperforms having access to all oracle documents by a noticeable margin, suggesting the necessity of multi-document retrieval to be another potential challenge for models.
>
> [1]  Hosseini et al. Efficient Solutions For An Intriguing Failure of LLMs: Long Context Window Does Not Mean LLMs Can Analyze Long Sequences Flawlessly. COLING 2025
>
> [2] Kurisinkel and Chen. LLM Based Multi-Document Summarization Exploiting Main-Event Biased Monotone Submodular Content Extraction. arXiv 2023
>
> [3] Tang et al. L-CiteEval: Do Long-Context Models Truly Leverage Context for Responding? arXiv 2024
>
> [4] Mullick et al. Long Dialog Summarization: An Analysis. arXiv 2024
>
> [5] Chen et al. LongLeader: A Comprehensive Leaderboard for Large Language Models in Long-context Scenarios. NAACL 2025

---

> > ### Comment · Reviewer_Witb · 2025-06-10
> >
> > Thanks for the insights, please include these in the main paper or appendix as these would be very insightful for future readers of this work

---

### Official Review · Reviewer_aSkS · 2025-05-13

**Rating:** 8
**Confidence:** 4
**Ethics Flag:** 1

**Summary:**

The paper presents 2 benchmarks for open-domain multi-document summarization (ODMDS). It does so by modifying two query-focused  multi-document summarization (QMDS) and reframing them as ODMDS:
1. ODSum-Story: The book-length query-focused summarization dataset SQuaLITY is converted to an ODMDS problem by chunking the books based on chapters and aggregating the chapters from different books in the dataset. The queries are decontextualized, and hints are added to enable the retrieval of the correct books.
2. ODSum-Meet: Similarly, the dataset of meeting transcript QA QMSum is transformed into ODMDS. The meetings have similar themes. So queries are clustered by an embedding model. Next, the query cluster is merged into one target query by an LLM. The corresponding summary for every query in the cluster is combined to form the target summary.

The authors manually verify the integrity of their post-processing by sampling 40 queries each from the two benchmarks and annotating them for (1) coherence and fluency, (2) need for open-domain retrieval, (3) need for multi-document aggregation, (4) query-summary relevance and coverage, and (5) factual consistency and accuracy. On all 5 metrics, more than 85-90% of the samples achieve a Likert score >= 4 out of 5.

The authors evaluate several retrievers (sparse, dense) with/without a reranker for information retrieval metrics on the benchmark. Similarly, several LLMs are evaluated for end-to-end retrieve-and-read performance paired with the different retrievers. Summary quality is measured by ROUGE and automated LM scoring (G-Eval) against the reference summary.

Results show that there is significant room for improvement in all stages: retrievers and summary writers. The performance with the Oracle documents is also low (70% on ODSum-Story and 54% on ODSum-Meet). The need for any retrieval, and in particular the retrieval of multiple documents, is established by ablations.

---

After Author Rebuttal (6/18)
---

I may be late in updating my review, but I feel my queries were sufficiently answered. Raising the score from 6 to 8.

**Questions To Authors:**

1. How are meeting transcripts in QMSum divided into documents?
2. How are gold documents annotated for the IR metrics? For SQuaLITY, is any chapter from the correct book scored as correct? For QMSum is any segment from the relevant meetings to the query cluster counted as positive?
3. App B: Transforming QMSum: Doesn't the summary merging step implicitly require the LLM to be excellent at QMDS? This requirement seems to be at odds with the low G-Eval scores of the LLMs on ODSum-Meet with Oracle documents. How do you explain the discrepancy?
4. App F: Human Validation: Did the annotators thoroughly verify Multi-document Information Aggregation by going back and forth between the summary and the reference documents?
5. App F: Human Validation: In both of these benchmarks, it is odd to require "open-domain" retrieval since the source tasks relied on one book/meeting transcript. In strict terms, is this checking whether a long context is required as opposed to a single paragraph?

Minor Queries
---
1. Sec 3.2: ODSum-Story construction: Some queries in SQuaLITY (especially the example shown on line 164) seem to ask broad questions about the book such as asking for a summary. Doesn't decontextualizing the query contaminate these examples?
2. Fig 1: The arrow labeled "Decontextualize" points to the summary instead of the query.
3. Do you provide all relevant documents in the Oracle setting or the top-8/top-3 documents in the two benchmarks?

**Reasons To Accept:**

- The benchmark makes a best-effort to source a realistic open-domain multi-document summarization task.
    - The authors establish that the systems actually benefit from aggregating multiple documents
    - The target summaries are sourced from previous carefully constructed datasets
    - This work identifies a missing benchmark in the literature and fills the gap
- Ablations are designed to prove that the task cannot be "gamed" by no-context and single-document baselines

**Reasons To Reject:**

- The large headroom in the improvement of response quality with Oracle documents is concerning
    - Even with Oracle documents, the best LLMs benchmarked achieve a G-Eval score of 70% on ODSum-Story and 54% on ODSum-Meet
    - The source of the remaining errors is concerning because it could point to (1) a low upper-bound score even with better retrievers, (2) it could point to an issue with the G-Eval metric and its use in this benchmark, (3) errors in the summary aggregation process for ODSum-Meet
    - Some analysis of the remaining error will be useful to rule out potential issues with the dataset creation and metrics

---

> ### Author Response · Authors · 2025-06-03
> **Rebuttal**
>
> Thank you for reviewing our work! We would like to address your concerns and questions as follows:
>
> > Q1: The large headroom in the improvement of response quality with Oracle documents is concerning
>
> Thanks for your suggestions! We conducted a small-scale error analysis study with the top-performing models in Tables 6 and 7 for the oracle setting. These models were DeepSeek-V3 for ODSum-Story (G-Eval score of 69.85) and GPT-4o for ODSum-Meet (G-Eval score of 53.67).
>
> For this study, forty summaries—twenty per subset—were randomly sampled from the pools of summaries generated by DeepSeek-V3 given the ODSum-Story oracle documents and GPT-4o given the ODSum-Meet oracle documents. The evaluations were divided up equally among two annotators, who selected all the error type labels that applied to each generated summary given the query, the gold-standard summary, and the oracle documents as reference. The table below depicts each error type label along with the prevalence of that error for the two subsets.
>
> Error Type  | Error Prevalence in ODSum-Story Oracle-DeepSeek-V3 Subset (N = 20) | Error Prevalence in ODSum-Meet Oracle-GPT-4o Subset (N = 20)
> -|-|-
> Missing many minor, fine details | 40% | 60%
> Missing many critical, important details | 0% | 35%
> Minor query misunderstanding | 0% | 5%
> Major query misunderstanding | 0% | 10%
> Presence of hallucinations | 10% | 10%
> Excessively general or vague response | 0% | 10%
>
>
> > The source of the remaining errors is concerning because it could point to (1) a low upper-bound score even with better retrievers
>
> Our error analysis study indicates that the generated summaries in the subsets often lacked many minor details (40% for ODSum-Story and 60% for ODSum-Meet), with ODSum-Meet also lacking many major details (35% of the time) and additional error types appearing less frequently. These initial findings are consistent with works [1] [2] [3] that demonstrate that even frontier LLMs may still find multi-document tasks difficult due to limitations in multi-document abilities. Furthermore, our human evaluation found ODSum-Meet to be more difficult than ODSum-Story, which aligns with the delta in G-Eval scores across the subsets (53.67 vs. 69.85). This observation is also supported by the original QMSum benchmark paper and recent works [4] [5] performing evaluations upon QMSum that attest to this benchmark’s difficulty.
>
> > (2) it could point to an issue with the G-Eval metric and its use in this benchmark,
>
> In our study, we observed trends where the examples for which the annotators did not observe any errors had higher G-Eval scores. Since our error analysis study here is limited by its small scale, we will scale up the study for the revised manuscript, with the addition of annotators evaluating the reliability and consistency of the G-Eval metric and computing its correlation with human evaluations.
>
> > (3) errors in the summary aggregation process for ODSum-Meet
>
> The summary aggregation process during the ODSum-Meet benchmark construction described in Appendix B involved the usage of an LLM as solely a tool to combine the multiple smaller gold-standard summaries for the queries being merged together. We do not anticipate errors from this process given the restricted scope of the LLM’s task, and further confirm the validity of the aggregated summaries in our dataset quality human evaluation study in Table 3.
>
> > Some analysis of the remaining error will be useful to rule out potential issues with the dataset creation and metrics
>
> We refer to the existing results in our paper to help analyze the remaining error beyond the oracle setting. As shown in Tables 6 and 7, the summarization performance of models, even for advanced generators, is heavily dependent on the quality of the retrieved documents. Additionally, Table 8 in the appendix illustrates that even having access to the most relevant oracle document underperforms having access to all oracle documents by a noticeable margin. This means that the inability of retrievers to fetch more than one relevant document—especially in ODSum-Meet, which had ~30% recall during retrieval (Table 4) out of an average of 3.04 oracle documents per query (Appendix E.1), resulting in ~1 retrieved document on average—would also negatively impact downstream generation performance.
>
> [1]  Hosseini et al. Efficient Solutions For An Intriguing Failure of LLMs: Long Context Window Does Not Mean LLMs Can Analyze Long Sequences Flawlessly. COLING 2025
>
> [2] Kurisinkel and Chen. LLM Based Multi-Document Summarization Exploiting Main-Event Biased Monotone Submodular Content Extraction. arXiv 2023
>
> [3] Tang et al. L-CiteEval: Do Long-Context Models Truly Leverage Context for Responding? arXiv 2024
>
> [4] Mullick et al. Long Dialog Summarization: An Analysis. arXiv 2024
>
> [5] Chen et al. LongLeader: A Comprehensive Leaderboard for Large Language Models in Long-context Scenarios. NAACL 2025

---

> > ### Author Response · Authors · 2025-06-03
> > **Rebuttal Continued**
> >
> > We thank the reviewer for all of their insightful questions, and continue our rebuttal response in this comment.
> >
> > > How are meeting transcripts in QMSum divided into documents?
> >
> > The meeting transcripts in QMSum were taken as whole without modification when collecting the documents for ODSum-Meet.
> >
> > > How are gold documents annotated for the IR metrics? For SQuaLITY, is any chapter from the correct book scored as correct? For QMSum is any segment from the relevant meetings to the query cluster counted as positive?
> >
> > Yes, ODSum-Story includes all chapters from the correct book as gold documents, and ODSum-Meet collects all meeting transcripts relevant to queries in the cluster as gold documents.
> >
> > > App B: Transforming QMSum: Doesn't the summary merging step implicitly require the LLM to be excellent at QMDS? This requirement seems to be at odds with the low G-Eval scores of the LLMs on ODSum-Meet with Oracle documents. How do you explain the discrepancy?
> >
> > Similar queries in QMSum and their associated summaries were clustered via text embedding retrieval. LLMs were only used as tools (no knowledge of QMDS required) to combine the short summaries together to form the merged summary (more details in Appendix B).
> >
> > > App F: Human Validation: Did the annotators thoroughly verify Multi-document Information Aggregation by going back and forth between the summary and the reference documents?
> >
> > Yes, annotators scanned the reference documents for information contained in the summaries and confirmed whether the information was spread out among multiple documents, going back and forth as needed.
> >
> > > App F: Human Validation: In both of these benchmarks, it is odd to require "open-domain" retrieval since the source tasks relied on one book/meeting transcript. In strict terms, is this checking whether a long context is required as opposed to a single paragraph?
> >
> > As each chapter and meeting transcript is treated as a separate document, multiple documents must still be retrieved. This is what the "Open-domain Retrieval Necessity" criterion was designed to validate.
> >
> > *Minor queries:*
> >
> > > Sec 3.2: ODSum-Story construction: Some queries in SQuaLITY (especially the example shown on line 164) seem to ask broad questions about the book such as asking for a summary. Doesn't decontextualizing the query contaminate these examples?
> >
> > During the generation process for decontextualized queries, a human annotator was in the loop as described on line 160 to ensure that the generated queries only added generic or vague context without including any specific details that would reveal the answer. The highly detailed nature of the gold-standard summaries in ODSum-Story necessitated that models retrieve information beyond that in the decontextualized query. For example, the query from line 164 lacks all of the specific details from the first sentence of its gold-standard summary—"The story takes place in Dynamopolis, a city in North America, in the year 819"—and from the rest of the summary.
> >
> > > Fig 1: The arrow labeled "Decontextualize" points to the summary instead of the query.
> >
> > Thank you for this catch! We will update Figure 1 to incorporate this fix in the revised manuscript.
> >
> > > Do you provide all relevant documents in the Oracle setting or the top-8/top-3 documents in the two benchmarks?
> >
> > Yes, all relevant documents were provided in the Oracle setting.

---

> > ### Comment · Reviewer_aSkS · 2025-06-08
> > **Re: Author response**
> >
> > Thank for the follow-up analysis. The results are very encouraging i.e. the low scores with Oracle documents point to poor summarizer performance. As such, there is large scope for improvement of the QMDS system in addition to improvement in retrieval.
> >
> > Follow-up questions:
> > 1. Is it fair to say that the remaining room for improvement in the Oracle setting is due to (1) known issues with handling very long context (entire book, multiple meetings) and (2) stylistic issues of what the LLM prioritizes to retain in the summary vs a human annotator?
> > 2. Because oracle documents are defined very loosely, a retriever with 100% recall will necessarily retrieve a long context. Am I getting that right?

---

> > > ### Author Response · Authors · 2025-06-10
> > > **Response to Reviewer aSkS's Follow-up Questions**
> > >
> > > Thank you for the follow-up questions!
> > >
> > > > Is it fair to say that the remaining room for improvement in the Oracle setting is due to (1) known issues with handling very long context (entire book, multiple meetings) and (2) stylistic issues of what the LLM prioritizes to retain in the summary vs a human annotator?
> > >
> > > Yes, both of your points capture key challenges faced in the Oracle setting. For ODSum-Story, the average query has 8.79 gold documents (Appendix E.1) with an average length of 845.44 tokens (Table 2), yielding an average context length of ~7.5k tokens. For ODSum-Meet, the average query has 3.04 gold documents with an average length of 7207.27 tokens, leading to an average context length of ~22k tokens. Processing contexts of this length, containing multiple book chapters or meeting transcripts, remains challenging for models due to known limitations in handling long contexts.
> > >
> > > Additionally, stylistic issues in what LLMs prioritize in the summary compared to human annotators add further difficulty to this task. In our small-scale error analysis, we found that nearly half of the generated summaries in the ODSum-Story Oracle-DeepSeek-V3 subset omitted many minor but relevant details. The gold summaries in SQuALITY, written by expert annotators, incorporate many specific details from the stories, which can be difficult for LLMs that favor more generic summarization. We observed a similar pattern in the ODSum-Meet Oracle-GPT-4o subset, where many generated summaries lacked both minor and critical details. This may stem from the fact that ODSum-Meet aggregates multiple QMSum queries and their corresponding gold summaries into a single merged query-summary pair. Because the merged gold summary is constructed to provide balanced coverage across all sub-queries, LLMs face a stylistic challenge: their generated summaries may prioritize some parts over others, rather than covering all parts of the query in depth.
> > >
> > > > Because oracle documents are defined very loosely, a retriever with 100% recall will necessarily retrieve a long context. Am I getting that right?
> > >
> > > Thank you for the comment. As you correctly pointed out, achieving 100% recall requires retrieving multiple long-form documents, resulting in a long input context for the LLM generator.
> > > The trade-off is that a more precise retriever that selects only the most relevant segments could produce shorter, more focused contexts, which may help the generator perform better. At the same time, many questions in our dataset require integrating information spread across many places in the original document. This makes it non-trivial to pinpoint a minimal set of gold segments, and motivates our broader oracle definition.
> > >
> > > Our goal is to highlight the challenges of both retrieval and long-context generation. Our experiments show that even under a high-recall oracle setting, long-context processing remains difficult, especially when many loosely related segments must be considered. We will add further clarification to the paper on this point.

---

### Decision · Program_Chairs · 2025-07-08

**Decision:**

Accept

**Comment:**

This paper creates two benchmarks for query-focused  multi-document summarization by modifying existing single-document summarization resources and repurposing them as multi-document ones.

    ODSum-Story: The book-length query-focused summarization dataset SQuaLITY is converted to a multi-document dataset by chunking the books based on chapters and aggregating the chapters from different books in the dataset. The queries are decontextualized, and hints are added to enable the retrieval of the correct books.

    ODSum-Meet: the dataset of meeting transcript QA QMSum is transformed, exploiting the fact that meetings have similar themes. So queries are clustered by an embedding model and clusters are  merged into one target query by an LLM. The corresponding summary for every query in the cluster is combined to form the target summary.

The authors manually verify the integrity of their post-processing by sampling 40 queries each from the two benchmarks and annotating them for (1) coherence and fluency, (2) need for open-domain retrieval, (3) need for multi-document aggregation, (4) query-summary relevance and coverage, and (5) factual consistency and accuracy.

The authors evaluate several retrievers (sparse, dense) with/without a reranker for information retrieval metrics on the benchmark. Similarly, several LLMs are evaluated for end-to-end retrieve-and-read performance paired with the different retrievers. Summary quality is measured by ROUGE and automated LM scoring (G-Eval) against the reference summary.

Results show that there is significant room for improvement in all stages: retrievers and summary writers. The performance with the Oracle documents is also low (70% on ODSum-Story and 54% on ODSum-Meet). The need for any retrieval, and in particular the retrieval of multiple documents, is established by ablations.

There is a lack of resources for query-focused multi-document summarization in the literature, and the paper aims to fill this gap. The authors have responded to the reviewer's queries and added error analysis results during the rebuttal. I would be interested to see what happens with a pattern matching method that retrieves sentences which match specific terms from the query (e.g., named entities such as "Christopher" or terms such as "remote control"). And if no retrieval takes place but the documents are given as input, in as much as they fit the context length.